# A rapid-application emissions-to-impacts tool for scenario assessment: Probabilistic Regional Impacts from Model patterns and Emissions (PRIME)

**Camilla Mathison**[1,3], **Eleanor J. Burke**[1], **Gregory Munday**[1], **Chris D. Jones**[1,8], **Chris J. Smith**[4], **Norman J. Steinert**[5,6], **Andy J. Wiltshire**[1,7], **Chris Huntingford**[2], **Eszter Kovacs**[3], **Laila K. Gohar**[1], **Rebecca M. Varney**[7], and **Douglas McNeall**[1,7]

[1]Met Office Hadley Centre, Exeter, UK
[2]UK Centre for Ecology and Hydrology, Wallingford, UK
[3]School of Geography, University of Leeds, Leeds, UK
[4]International Institute for Applied Systems Analysis (IIASA), Laxenburg, Austria
[5]Norwegian Research Centre AS (NORCE), Bjerknes Centre for Climate Research, Bergen, Norway
[6]CICERO Center for International Climate Research, Oslo, Norway
[7]Faculty of Environment, Science and Economy, University of Exeter, Exeter, UK
[8]School of Geographical Sciences, University of Bristol, Bristol, UK

**Correspondence:** Camilla Mathison (camilla.mathison@metoffice.gov.uk)

**Abstract.** `TS1` `TS2` Climate policies evolve quickly, and new scenarios designed around these policies are used to illustrate how they impact global mean temperatures using simple climate models (or climate emulators). Simple climate models are extremely efficient, although some can only provide global estimates of climate metrics such as mean surface temperature, $CO_2$ concentration and effective radiative forcing. Within the Intergovernmental Panel on Climate Change (IPCC) framework, understanding of the regional impacts of scenarios that include the most recent science is needed to allow targeted policy decisions to be made quickly. To address this, we present PRIME (Probabilistic Regional Impacts from Model patterns and Emissions), a new flexible probabilistic framework which aims to provide an efficient mechanism to run new scenarios without the significant overheads of larger, more complex Earth system models (ESMs). PRIME provides the capability to include features of the most recent ESM projections, science and scenarios to run ensemble simulations on multi-centennial timescales and include analyses of many key variables that are relevant and important for impact assessments. We use a simple climate model to provide the global temperature response to emissions scenarios. These estimated temperatures are used to scale monthly mean patterns from a large number of CMIP6 ESMs. These patterns provide the inputs to a "weather generator" algorithm and a land surface model. The PRIME system thus generates an end-to-end estimate of the land surface impacts from the emissions scenarios. We test PRIME using known scenarios in the form of the shared socioeconomic pathways (SSPs), to demonstrate that our model reproduces the ESM climate responses to these scenarios. We show results for a range of scenarios: the SSP5–8.5 high-emissions scenario was used to define the patterns, and SSP1–2.6, a mitigation scenario with low emissions, and SSP5–3.4-OS, an overshoot scenario, were used as verification data. PRIME correctly represents the climate response (and spread) for these known scenarios, which gives us confidence our simulation framework will be useful for rapidly providing probabilistic spatially resolved information for novel climate scenarios, thereby substantially reducing the time between new scenarios being released and the availability of regional impact information.

# 1   Introduction

A major gap currently exists in our capability to rapidly assess and predict regional impacts of climate change in response to novel future pathways of climate change and rapidly evolving policies. Sophisticated and specialist climate impact models exist that assess the regional implications of future climate scenarios for a range of impact sectors, such as crops, biomes, water, fire and permafrost, for example through the Inter-Sectoral Impact Model Intercomparison Project (ISIMIP; Frieler et al., 2017; Warszawski et al., 2014, 2013). ISIMIP provides a consistent framework for assessing climate impacts using a large ensemble of models across a range of sectors. However, impact models are often specific to particular sectors and are in themselves complicated to set up. Usually, their use occurs at the end of a long chain of events: commencing with generation of emissions scenarios, running one or more Earth system models (ESMs), potentially bias-correcting ESM output and then finally running the impact model.

In order to assess impacts resulting from climate change more systematically, ISIMIP provides output of ESMs to impact modellers. But even then, there is a long delay from creation of the scenarios to our ability to assess their impacts. For example, the most recent impact assessment of the Intergovernmental Panel on Climate Change (IPCC), the Sixth Assessment Report (AR6) Working Group II (Climate Change Impacts, Adaptation and Vulnerability: WGII) (IPCC, 2022b), relies heavily on literature based on impact studies using output from the previous generation of the Coupled Model Intercomparison Project (CMIP5), rather than the most recent CMIP6 used in AR6 Working Group I (IPCC, 2021). This means that both the scenarios (Representative Concentration Pathways; van Vuuren et al., 2011) and climate models themselves (e.g. HadGEM2-ES Jones et al., 2011; Collins et al., 2011) used to assess climate impacts by the IPCC are at least a decade old.

This apparent bottleneck is caused by the significant issue that ESMs, which are the main mechanism for projecting future climate change, are computationally demanding, so only a limited number of simulations may be performed. As ESMs take years to develop, test and run, scenarios of future climate change are only produced periodically on a time frame designed to align with IPCC assessment reports, such as contributions to the CMIP phases (Taylor et al., 2012; Eyring et al., 2016). Nevertheless, ESMs remain the best tools for understanding mechanisms of climate change, and regional climate projections could not be performed without them.

One popular method to enable projections of future climate change for novel emissions scenarios, and yet capture the process understanding implicit in the ESMs simulations that do exist, is via "pattern scaling" (Zelazowski et al., 2018; James et al., 2017; Huntingford et al., 2010; Mitchell, 2003). Such scaling assumes that local and monthly changes in near-surface meteorological conditions correlate linearly with the level of global warming. Lee et al. (2021) note that pattern scaling has known limitations, for example having lower skill for variables with large spatial variability (Herger et al., 2015; Tebaldi and Arblaster, 2014) or when attempting to recreate moving boundaries such as sea ice extent and snow cover (Collins et al., 2013). Nonetheless, the benefits of pattern scaling to enable rapid reconstruction of spatial patterns based on global temperature make it an extremely valuable tool for studying, for example, carbon cycle feedbacks using an intermediate-complexity climate model (Mercado et al., 2009; Burke et al., 2017).

Other tools are currently being developed to explore the use of pattern scaling for local climate change impacts. The Modular Earth System Model Emulator with spatially resolved output (MESMER; Beusch et al., 2020) draws on patterns of temperature from CMIP6 models, and its extension to this (MESMER-M; Nath et al., 2022) focuses on spatially resolved monthly temperature or extremes (MESMER-X; Quilcaille et al., 2022). MESMER is an emulator of temperature patterns and uses a stochastic representation of natural variability. Goodwin et al. (2020) have also used pattern scaling with the WASP global emulator to look at local temperature projections. Alternatively, the STITCHES system (Tebaldi et al., 2022) presents an option for ESM emulation for impact research by "stitching" together ESM output from known scenarios, and ClimateBench v1.0 (Watson-Parris et al., 2022) benchmarks machine learning emulators that predict annual mean global distributions of temperature, diurnal temperature range and precipitation.

We use pattern-scaled climate variables instead of ESM output to drive our impact model, because this approach offers a useful opportunity to more quickly derive impact information from new scenarios. However, this does not imply that pattern-scaled climate variables should replace ESMs or ISIMIP bias-corrected data but could provide a steer on which scenarios would be most useful for ESMs to run or which ones to bias-correct for use in more specialist impact models. Global mean temperature is readily and efficiently calculated from emissions scenarios using one of a range of climate emulators (Nicholls et al., 2020), which are computationally cheap to run. The regional climate patterns are then scaled by applying global mean temperatures produced from emulators. The ability to run simulations without running the full ESM is particularly useful for assessing novel scenarios, particularly those that are regularly updated (Richters et al., 2022) to address specific questions around Paris Agreement

compliance and overshoot (Rogelj et al., 2018; IPCC, 2022a) or to answer "what-if" questions relating to the Earth's geophysical response (Dvorak et al., 2022). These scenarios may never be run through full ESMs because of the vast computational resources required, but understanding their regional impacts may be important in answering adaptation and mitigation questions. The efficiency and flexibility of emulators allow them to run ensembles in a probabilistic Monte Carlo framework, spanning the range of assessed climate uncertainty with different parameter choices (Nicholls et al., 2021). We propose that these emulator systems provide an important and relatively (computationally) cheap first look at new scenarios that could inform future ESM developments.

Here we present PRIME (Probabilistic Regional Impacts from Model patterns and Emissions): a framework designed to bridge the gap between scenarios and impacts in a computationally efficient manner. PRIME builds on previous work of Huntingford and Cox (2000) which culminated in the formal coupling of the analogue model (i.e. energy balance model or EBM plus climate patterns) to a vegetation model that created the modelling framework called IMOGEN (Integrated Model Of Global Effects of climatic aNomalies – Huntingford et al., 2010). IMOGEN also contains a simple single box representation of the oceanic drawdown of atmospheric carbon dioxide as a function of global mean temperature change over the oceans and $CO_2$ level. As such, IMOGEN contains a global carbon cycle and so instead may be forced by $CO_2$ emissions, and from this by accounting for land–atmosphere and land–ocean interactions, atmospheric $CO_2$ levels are projected. IMOGEN was originally calibrated against ESMs in the Coupled Model Intercomparison Project version 3 (CMIP3) (Zelazowski et al., 2018) and later against version 5 (CMIP5). In this paper we replace the EBM in IMOGEN with the FaIR model, which means we can extend beyond the influence of $CO_2$ and consider other greenhouse gases and short-lived climate forcers that also influence the global temperature. Using FaIR also brings in the latest science from the reduced-complexity modelling community. While the underlying IMOGEN model remains inspired by and based largely on the code in IMOGEN (Huntingford and Cox, 2000), in PRIME we update the patterns to use those from CMIP6 models and couple the output from IMOGEN to a full land surface model to study land-based impacts.

Our approach combines the full range of FaIR temperature responses with the full range of CMIP ESM patterns. We note a pattern effect relating warming to climate sensitivity has been shown in the literature (Andrews and Webb, 2018; Ringer et al., 2014). However, assessments of simulated impacts in the CMIP6 ensemble sampling a wide range of impact metrics from multiple regions found little or no correlation with climate sensitivity for most regions and climate drivers (Swaminathan et al., 2024), which contributes to justifying the approach to treat these independently. Other studies have found changes to circulation patterns and dynami-

cal regimes more important for climate patterns than global-scale thermodynamical response (Ribes et al., 2021, 2022; Palmer et al., 2023). To maximise our sampling of uncertainty, we therefore take the pragmatic decision to co-vary all patterns with sampled temperature pathways.

In this way, PRIME facilitates faster pull-through of state-of-the-art science from the latest scenarios and regional climate change patterns (from the latest ESMs) all the way to the simulation of regional impacts. PRIME includes the latest understanding of climate and carbon cycle feedbacks, the latest spatial patterns of climate change and a leading land surface model/impact model. In PRIME we accommodate a broad range of variables in addition to temperature, with a focus on those which are important for impact assessments. PRIME is a flexible framework, with ensemble members and patterns selected by the user, and is therefore dependent on their chosen application. However, we are developing software to simplify running the PRIME framework using the choices presented here, using Rose and Cylc (Oliver et al., 2018, 2024) – a group of utilities and specifications which provide a common way to manage the development and running of scientific application suites in both research and production environments. Rose and Cylc are used to ensure a consistent framework for managing and running meteorological and climate models; they are therefore ideally suited to this application. The elements of PRIME are explained in more detail in Sect. 2. An evaluation of the performance of the framework is provided in Sect. 3. Additional results that are relevant for impact applications are presented in Sect. 4, with discussion and conclusions in Sect. 5.

## 2  Methods

PRIME is a rapid-response tool designed to explore spatially resolved climate and impacts of scenarios as soon as they are developed. It draws on comprehensive CMIP multi-model ensemble results but extends these to fill gaps not yet populated by ESMs or impact models and can extend simulations into the future to simulate multi-century responses. PRIME produces probabilistic sampling of a range of uncertainties, including global climate and carbon cycle sensitivity and spatial patterns of climate change. It opens the potential to also span perturbed parameter uncertainty in land and impact models and provides the ability to propagate constraints onto impact projections through either prior constraint on parameters or posterior selection of ensemble members.

Figure 1 shows the components that make up the PRIME framework. The starting point is emissions scenarios such as from integrated assessment models (IAMs), which are used to drive the global climate emulator FaIR v1.6.2 (Smith et al., 2018, see Sect. 2.1). FaIR can probabilistically sample uncertainty in climate and carbon cycle response to emissions. Its global mean temperature projections are then used to reconstruct the regional climate change for a number of cli-

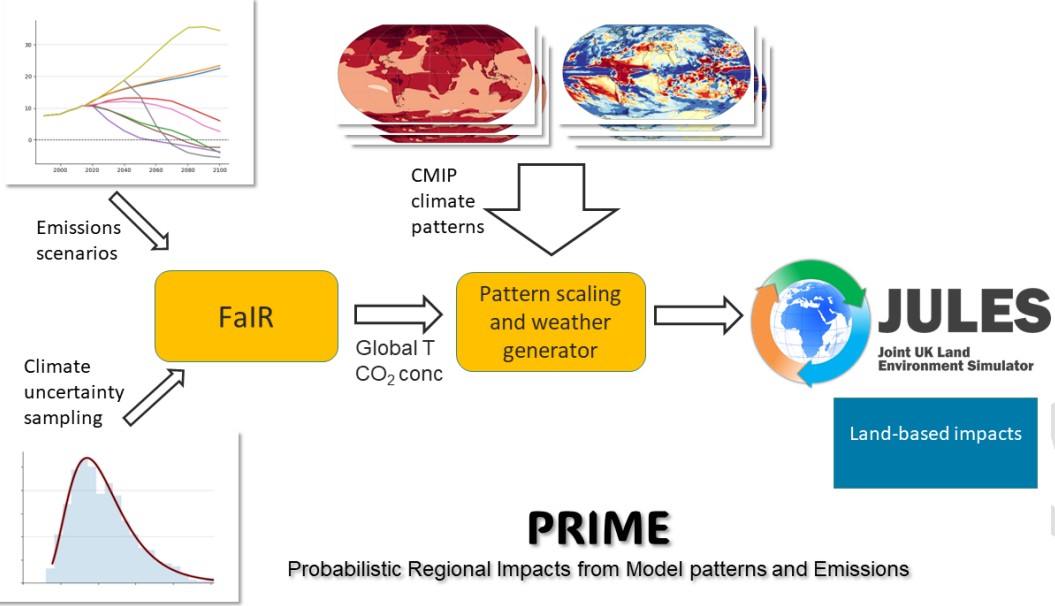

**Figure 1.** Schematic of the PRIME framework. Emissions scenarios provide input in terms of emissions of $CO_2$, other greenhouse gases and aerosols and can be taken from IAMs or idealised experiments. The FaIR climate emulator samples uncertainty from the climate and carbon cycle response to emissions and outputs global temperature and $CO_2$ concentration. PRIME then scales patterns of climate change from CMIP climate models by the global temperature and uses a weather generator to downscale these to subdaily driving data for the JULES land surface model. JULES outputs a broad range of land-based impact-relevant quantities such as gross primary productivity (GPP), net primary productivity (NPP), vegetation cover, soil moisture and runoff.

mate variables using the patterns derived from ESMs (see Sect. 2.2). These regional patterns, along with $CO_2$ concentrations from FaIR, are used to drive the JULES land surface model, from which various climate impacts can be derived.

## 2.1 Emulator of global temperature change

The Finite Amplitude Impulse Response (FaIR) model is a climate emulator that takes inputs of greenhouse gas and short-lived climate forcer emissions and produces projections of global mean surface temperature (Smith et al., 2018; Leach et al., 2021). FaIR calculates greenhouse gas concentrations (including $CO_2$) and effective radiative forcing as intermediate steps. FaIR contains modules that simulate the carbon cycle feedback (changes in uptake of $CO_2$ by land and ocean sinks with increasing $CO_2$ emissions and warming) and forcing from aerosols, ozone, land use change and several other categories of anthropogenic and natural forcings. These relationships in FaIR are designed to capture the large-scale behaviour of complex Earth system models and are governed by a number of tuneable parameters.

As part of the IPCC AR6 Working Group 1 (The Physical Science Basis of Climate Change: WGI; IPCC, 2021), a 1.6-million-member prior ensemble of FaIR v1.6.2 was produced. This large ensemble is reduced using the historical temperature record to eliminate those members with a large error, with the aim of reproducing the uncertainty range in

present day relative to the pre-industrial period. Through simultaneously constraining on several observable and emergent climate metrics including equilibrium climate sensitivity (ECS), transient climate response (TCR), aerosol, $CO_2$ concentration and ocean heat change, the ensemble is reduced to the 2237 members used in AR6 (Forster et al., 2021; Smith et al., 2021, 2023). This ensemble of 2237 parameter sets was taken forward and used for assessing emissions pathways derived from IAMs in the IPCC AR6 Working Group III (Mitigation of Climate Change) report (Riahi et al., 2022). It is this ensemble (Smith, 2022) we select from this version of the PRIME framework. This AR6 calibration is described in detail in fair-calibrate V1.4.0 (Smith et al., 2024), which shows the range of climate uncertainty parameters sampled; these include radiative forcing from different drivers (including aerosols), carbon cycle sensitivities, timescales of climate response to forcing and climate sensitivity. In this study, to make the ensemble size manageable, we reduce the total number of ensemble members by subsampling from within the 2237 parameter sets to explore the full range of global temperature sensitivity using several percentiles at 0 %, 1 %, 5 %, 25 %, 50 %, 75 %, 95 %, 99 % and 100 %; these are selected using one scenario so that scenarios can be compared against each other. We use a single scenario to define the ensemble member per percentile because each scenario will have different ensemble members for each percentile. For example, the 50th percentile ensemble member

for SSP5–8.5 would not be the same ensemble member as the 50th percentile for SSP5–3.4-OS. We choose the same ensemble members for all scenarios to make the comparison between scenarios easier. In this framework we also output $CO_2$ concentrations from FaIR for use in JULES; in future work we intend to explore the selection of ensemble members based on sampling the $CO_2$ range of uncertainty as well as temperature, but this is not explored here. In this paper, we present just one way that the user can choose to run PRIME, but these choices are not intrinsic to PRIME as a framework. The optimal sampling strategy within the distribution of FaIR outputs and climate patterns (see Sect. 2.2) can and will vary on a case-by-case basis depending on the desired use of the framework. Additionally, all the percentiles are available from FaIR if a user chooses to use them.

## 2.2 Spatial patterns and temporal downscaling of climate change

In PRIME, we use an early version of pattern scaling developed by Huntingford and Cox (2000). We derive relationships for eight climate variables (near-surface air temperature, diurnal temperature range, precipitation, shortwave radiation, longwave radiation, near-surface specific humidity, 10 m wind speed and surface pressure) at each grid cell, by a linear regression between global mean temperature change and anomalies relative to the 1850–1889 mean for all climate variables with an intercept set to zero. Monthly patterns for each of 34 CMIP6 models (see Table S1 in the Supplement for the models and realisations used) are calculated using the SSP5–8.5 emissions scenario: sampling the CMIP6 ensemble's range of uncertainty. We generate the patterns separately for each CMIP model (using the recipe available in ESMValTool; see the data availability section); the regression is calculated with points from the duration of SSP5–8.5, from 2015–2100. This means that PRIME is run for each CMIP pattern individually (we do not run it using the average CMIP pattern). This use of a large proportion of the CMIP6 ensemble means that PRIME considers many combinations of ESM output for a broad range of climates represented by the CMIP6 ensemble. Wells et al. (2023) show that whilst selecting patterns derived from emissions scenarios with radiative forcings closer to the target scenario results in the lowest emulation errors, the best all-round performance is obtained by using a high warming scenario to obtain the patterns, hence our choice of SSP5–8.5 as our training scenario. For further details on the patterns evaluation, see Sect. 3.2.

Within PRIME we use patterns for all input variables required to run the JULES land surface model. JULES tends to be less sensitive to some of the input variables that do not typically scale as well with temperature, such as wind speed, pressure and longwave downwelling radiation, so we can include them without introducing erroneous output changes (see Sect. 3.2). It should be noted that we generate global patterns that include land and ocean, but in this analysis, we focus on the patterns over land for running JULES and considering land impacts. However, it would be possible also to use the patterns over the ocean for relevant downstream applications.

The spatial distribution of the monthly mean meteorology for each month of the transient simulation is reconstructed from the climate patterns multiplied by the global mean temperature change (Sect. 2.1) superimposed on an observed monthly climatology. This is done by IMOGEN (Huntingford et al., 2010). In this version of PRIME, the observed monthly climatology was constructed from the daily meteorological data provided by the GSWP3-W5E5 dataset from the ISIMIP3a project (Frieler et al., 2024) for the period 1901–1930. This was regridded to a resolution of N48 with a 3.75° longitude grid size and a 2.5° latitude grid size.

In addition, the weather generator in IMOGEN (Huntingford et al., 2010) is used to downscale the weather data from the monthly to hourly time step, which is the temporal resolution used to drive JULES. This method is described in detail in Mathison et al. (2023). One limitation of this method is the lack of variability in the driving humidity, temperature and radiation at both the subdaily and daily resolution. In the next version of PRIME, we will develop the temporal downscaling meteorology so that it coherently includes the effects of, for example, clouds on the diurnal cycle of the weather data.

The diurnal cycle in near-surface air temperature is defined using

$$T = T_o + \frac{\Delta T}{2} \cos\left(\frac{2\pi \left(t - t_{T_{\max}}\right)}{T_{\text{day}}}\right), \tag{1}$$

where $T_o$ and $\Delta T$ are the temperature and diurnal temperature ranges respectively. $T_{\text{day}}$ is the length of the day, i.e. 24 h; $t_{T_{\max}}$ is the time of day when the temperature is highest. $t_{T_{\max}}$ is calculated from the following equation, which assumes that it occurs 0.15 of the period between sunrise and sunset after solar noon:

$$t_{T_{\max}} = \frac{t_{\text{up}} + t_{\text{down}}}{2} + 0.15 \left(t_{\text{up}} - t_{\text{down}}\right), \tag{2}$$

where $t_{\text{up}}$ and $t_{\text{down}}$ are sunrise and sunset times. The downward shortwave radiation, which includes the diurnal cycle, is the daily mean downward shortwave radiation multiplied by a solar radiation normalisation factor, which depends on the position of the sun in the sky at each time step for each grid box. This means that subdaily downward shortwave radiation and temperature are estimated using these known factors and a sinusoidal function to represent the maximum and minimum daily range.

The downward longwave radiation has a dependence on temperature, $T$, that is an exponent power to 4, based on the theory of black-body radiation. However, if we assume the diurnal cycle in temperature is relatively small compared to background temperature, $T_0$ (at which longwave radiation is $R_{\text{lw},0}$), then we can linearise about these values. This gives

$$R_{lw} = R_{lw,0} \left( \frac{4T}{T_0} - 3 \right). \tag{3}$$

The IMOGEN (Huntingford et al., 2010) weather generator distributes monthly mean rainfall subject to a probability distribution that has fixed parameters in time (i.e. year), albeit dependent on month and location. For each year, a random number generator is applied to sample from the distribution. The distribution parameters are fitted to known historical gridded measurements of precipitation. Precipitation is split into three types: large-scale rain, convective rain and large-scale snow. These are considered to occur in a single event, with a globally specified duration parameter (6 h for convective rainfall, 1 h for large-scale rainfall and convective snowfall and large-scale snowfall). The type of precipitation at any particular time depends on the mean daily temperature. If the daily temperature is greater than 293.15 K, it is convective rain, between 275.15 and 293.15 K it is large-scale rain, and below 275.15 K it is large-scale snow. This precipitation is divided into events of randomly generated duration. If the maximum precipitation rate in any time step is greater than $350.0 \, \text{mm} \, \text{d}^{-1}$, the precipitation is again redistributed to reduce these values to less than the threshold.

## 2.3 Land surface and impact model

The Joint UK Land Environment Simulator Earth System (JULES; Best et al., 2011; Clark et al., 2011; Wiltshire et al., 2021) land surface model is a community model used both in a standalone model and as the land surface component of the UK Earth System Model (UKESM; Sellar et al., 2019). Here, JULES is used in standalone mode, driven by climate data reconstructed by combining the monthly patterns derived from the ESMs and the global mean temperature change from FaIR. The configuration of JULES used here is denoted JULES-ES (Mathison et al., 2023) and is the configuration used both in UKESM1 (Sellar et al., 2019) and to provide simulations for the Inter-Sectoral Impact Model Intercomparison Project (ISIMIP) in Mathison et al. (2023). In PRIME JULES-ES is also driven by the $CO_2$ concentration output from FaIR.

JULES-ES has nine natural plant functional types (PFTs; five types of trees, $C_3$ and $C_4$ grasses, and evergreen and deciduous shrubs) and four managed PFTs ($C_3$ and $C_4$ crop and pasture), where the managed PFTs are set to their observed values at 2005. The Top-down Representation of Interactive Foliage and Flora Including Dynamics (TRIFFID) dynamic vegetation model (Cox, 2001) determines the proportion of each PFT present in a grid cell. Nitrogen limitation on ecosystem carbon assimilation is represented in JULES-ES (Wiltshire et al., 2021). External nitrogen inputs are via biological nitrogen fixation and nitrogen deposition, and losses are via leaching and a gas loss term. Nitrogen limitation reduces the carbon use efficiency of the vegetation via a reduced net primary productivity and can slow soil decom-

position. The soil biogeochemistry is represented by a single bulk layer with four soil pools: two litter pools, a microbial biomass pool, and a humus pool each with an equivalent organic nitrogen pool. Inorganic nitrogen is converted from organic nitrogen and can be taken up by the plants.

## 3 PRIME evaluation

In this section, we evaluate PRIME. In this context, that means that we show that the framework is "fit for purpose" by testing it on scenarios where ESM simulations already exist. However, ultimately we want to use PRIME to produce land simulations for scenarios where ESMs have not been run. Here, we use CMIP6-simulated output for a range of different but well-known future climate scenarios: SSP1–2.6, SSP5–3.4-OS (these are verification scenarios) and SSP5–8.5 (this is the training scenario). We show that PRIME gives close agreement of global temperature and spatial patterns of climate, giving us confidence in its ability to be used to project as yet unsimulated scenarios. We also compare simulated land surface output from PRIME with that from CMIP6 for ESMs that have reported the required diagnostics.

PRIME has three distinct and independent steps, as described in Sect. 2: (i) time series of global temperature are produced from FaIR based on emissions of greenhouse gases and aerosols, (ii) spatial patterns of climate change are constructed from the global temperature based on CMIP simulations, and (iii) these climate patterns are used to drive JULES to simulate land surface outcomes. In this section, we present the evaluation of these three steps and at each step assess the agreement with existing output from CMIP6 or the IPCC AR6 assessment using various standard statistical methods. The chosen statistics vary with each step and include the mean absolute error (MAE), the root mean square error (RMSE), the Pearson correlation coefficient and the interquartile range (IQR) of model predictions.

## 3.1 Emulation of global temperature change

For the first time, IPCC AR6 was able to apply multiple lines of evidence to constrain future projections of global temperature from the CMIP6 ensemble. As such, the spread of global temperature in 2100 is smaller than if taken from raw CMIP6 ESM output (Lee et al., 2021). We compare the simulated global temperature from PRIME (run with emissions) with the constrained range assessed by IPCC (see Fig. 4.11 and Table 4.5 of Lee et al., 2021) in Fig. 2. The FaIR simulations used here do not include solar and volcanic fluctuations, instead focusing on the anthropogenic forcing, which is the main driver of human-induced effective radiative forcing and human-induced warming (Forster et al., 2023). The left panel shows the mean global mean surface air temperature (GSAT) from FaIR (solid lines) and 5th and 95th percentiles (shaded region). The right panel shows the mean and 5th–95th per-

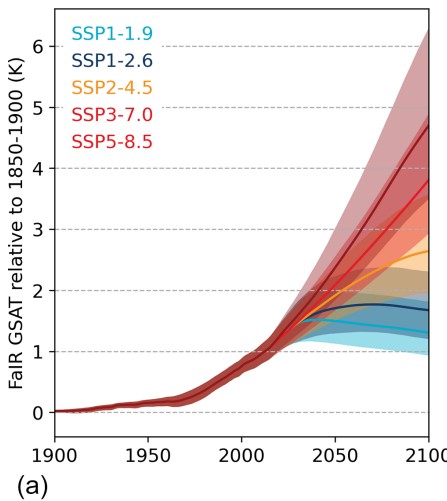
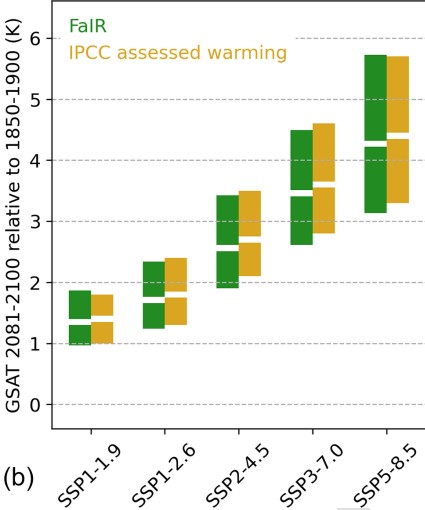

**Figure 2. (a)** Projected global temperature from FaIR for five SSP emissions scenarios and **(b)** comparison of end-of-century (2080–2100) mean warming from FaIR (green) and the IPCC AR6 assessment (yellow; Fig. 4.11 and Table 4.5 of Lee et al., 2021)

centile for the period 2090–2100 relative to 1850–1900 for each SSP scenario (we include the same scenarios here to enable comparison with the same plot in the IPCC report): SSP1–1.9, SSP1–2.6, SSP2–4.5, SSP3–7.0 and SSP5–8.5 for FaIR end-of-century GSAT values and the IPCC-constrained values. The end-of-century ranges in PRIME are close to the IPCC ranges with the time series and model spread consistent with the IPCC-constrained range.

The uncertainty in projected global mean temperature arises from uncertainty in both physical and biogeochemical feedbacks like the carbon cycle. We know atmospheric $CO_2$ is an additional direct driver of impacts; therefore this is another output from FaIR that is included in PRIME as an input to JULES-ES. Figure 3 shows the selection of ensemble members from the full FaIR distribution of 2237 members; nine percentiles (0 %, 1 %, 5 %, 25 %, 50 %, 75 %, 95 %, 99 % and 100 %) are chosen to explore the full range of global temperature sensitivity but make the data more manageable because they increase considerably when combined with the CMIP6 patterns (see Table S1 for a full list of those used) and run through JULES. PRIME samples the joint distribution of $CO_2$ and global temperature from the constrained FaIR ensemble. Figure 3 shows how the ensemble members selected span the distribution for SSP1–2.6 (similarly the joint distributions of $CO_2$ and temperature are also shown in Fig. S1 in the Supplement for SSP5–3.4-OS on the left and SSP5–8.5, the training scenario, on the right). As our primary aim is to sample future impacts associated with uncertain future temperature outcomes, we subsample the 2100 FaIR temperature distribution for the impact modelling. This results in a co-sampling of $CO_2$ levels that does not span the full uncertainty in resulting $CO_2$ concentrations. This is not a limitation of PRIME – other applications could use a different sampling strategy or use the full ensemble of 2237 mem-

bers. As mentioned in Sect. 2.1, the sampling strategy will depend on the intended application of the framework; the use of both temperature and $CO_2$ concentration from the FaIR distribution is discussed in Sect. 5.

## 3.2 Spatial patterns of climate change

Emulated climate patterns are evaluated against their CMIP6 equivalents for a number of scenarios. Alongside global comparison, four example regions are chosen to test the pattern evaluation at regional scales: the Amazon basin CE1, the Siberian forest, India and the United States of America. These regions span tropical and boreal ecosystems, temperate regions, and a region dominated by a monsoon climate. The climate patterns were evaluated against the out-of-sample CMIP6 runs of the SSP1–2.6 and SSP5–3.4-OS scenarios as SSP5–8.5 was used to train the pattern scaling.

Climatologies of each CMIP6 model were calculated by taking the mean of the period 1850–1889 inclusive. Anomalies were then calculated by subtracting the climatologies from the spatiotemporal CMIP6 datasets, ensuring that the variants of the historical runs matched those of the scenarios. To compare against these, predicted patterns for each ESM were compiled by multiplying annual mean GSAT data by pattern values at each grid point (see Methods, Sect. 2.2), creating a spatiotemporal dataset of anomalies for each variable (see Table 1). The predicted patterns were then evaluated against the anomaly datasets from CMIP6. The number of models included in the evaluation depends on the scenario, as not all CMIP6 models simulated every SSP. Here, 29 models are included in the evaluation against SSP1–2.6, and 15 are included in the evaluation of SSP5–3.4-OS, out of the 34 available model patterns.

**Table 1.** Summary table for evaluation section: root mean square error (labelled RMSE) between multi-model mean pattern predictions and CMIP6 both at mid-century (labelled "Mid", i.e. 2040–2060) and at the end of the century (labelled "End", i.e. 2080–2100) for values on land across the globe for each input variable and scenario. RMSE is spatially aggregated across months and models.

| Variable JULES inputs | Units | SSP1–2.6 RMSE | | SSP5–3.4-OS RMSE | | SSP5–8.5 RMSE | |
|---|---|---|---|---|---|---|---|
| | | Mid | End | Mid | End | Mid | End |
| Temperature | °C | 0.43 | 0.52 | 0.42 | 0.53 | 0.34 | 0.35 |
| Specific humidity | $g\,kg^{-1}$ | 0.0001 | 0.0002 | 0.0001 | 0.0002 | 0.0001 | 0.0001 |
| Precipitation | $mm\,d^{-1}$ | 0.20 | 0.22 | 0.22 | 0.22 | 0.18 | 0.17 |
| Wind | $m\,s^{-1}$ | 0.10 | 0.11 | 0.10 | 0.11 | 0.09 | 0.08 |
| Pressure | $kg\,m^{-1}\,s^{-2}$ | 32.11 | 36.79 | 36.29 | 34.53 | 28.55 | 25.81 |
| Shortwave radiation | $W\,m^{-2}$ | 1.75 | 2.16 | 1.85 | 2.19 | 1.88 | 1.98 |
| Longwave radiation | $W\,m^{-2}$ | 1.88 | 2.29 | 1.74 | 2.25 | 1.52 | 1.65 |
| Diurnal temperature range | °C | 0.12 | 0.13 | 0.09 | 0.11 | 0.09 | 0.09 |

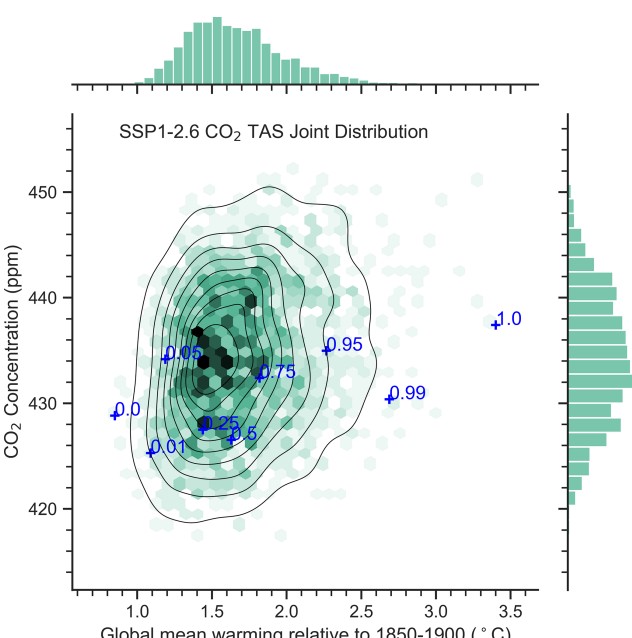

**Figure 3.** Joint frequency distribution from the FaIR simulations of global temperature rise and $CO_2$ concentration in 2100 for SSP1–2.6 emissions and the subselected percentiles (blue crosses) used to drive the JULES impact model. Shades of green denote the density of points with individual histograms above and to the right of the main panel. The 10 % confidence intervals are shown by the contours.

Our evaluation of the emulated patterns focuses on the ability to capture the mean and spread of the CMIP6 ensemble. The aim is for PRIME to appropriately capture the response to forcing across a range of scenarios and also the spatial uncertainty. We evaluate the pattern scaling by comparing mid-century (2040–2060 mean) and end-of-century (2080–2100 mean) predictions for all variables against CMIP6 anomalies for two out-of-sample scenarios – SSP1–2.6 and SSP5–3.4-OS – using pattern predicted ensemble means compared to the CMIP6 ensemble mean anomalies. In Fig. 4 we show evaluation of predictions of near-surface air temperature and precipitation. The right-hand column shows the error in our prediction of the CMIP6 multi-model mean. Temperature is clearly seen to scale well, with small errors during both the mid-century and the end of the century; however warming in the northern latitudes is generally slightly underestimated across both timescales. The results for precipitation show much more spatial variance. Patterns of change are well captured during both timescales, as can be seen in Fig. 4d, e, j and k, though prediction errors do occur for some regions (see Fig. 4f and l), for example in the Amazon, where rainfall is generally underestimated; southern Africa, where it is overestimated; and South-East Asia, where differences vary over timescales. We include further evaluation of patterns for the other JULES input variables in the Supplement (see Fig. S2 for specific humidity and wind, Fig. S3 for pressure and downwelling shortwave radiation, and Fig. S4 for downwelling longwave radiation and the diurnal temperature range). We also show all variables for SSP5–3.4-OS in the Supplement (these are shown in the same order with temperature and precipitation first in Figs. S5 to S8). We also show the training scenario, SSP5–8.5, in the Supplement for temperature and precipitation (see Fig. S9) as a sanity check. In general, errors in pressure (Fig. S3a–c and g–i), longwave downwelling radiation (Fig. S4a–c and g–i) and specific humidity (Fig. S2a–c and g–i) tend to be smaller, while errors in shortwave downwelling radiation (Fig. S3d–f and j–l),

wind (Fig. S2d–f and j–l) and the diurnal temperature range (Fig. S4d–f and j–l) tend to be larger, regardless of the scenario and particularly at the end of the century (Figs. S4l and S8l TS3). For downwelling shortwave radiation particularly, this is likely to be due to the influence of other factors such as aerosols.

In addition to evaluating the relative ensemble means, it is important to check that prediction errors fall within the range of responses seen in CMIP6. We therefore check that the absolute error in our predictions is small compared to the spread in the ensemble predictions. We calculate the absolute error of each model's prediction against its CMIP6 counterpart and take the mean over the ensemble at each grid point (Fig. 5, central column). If the mean absolute error (MAE) is low relative to the CMIP6 IQR (Fig. 5, left column), this suggests the pattern-scaling technique is not adding significant variation in its predictions beyond that driven by the differences between the patterns. We can therefore be confident that despite deficiencies, the ensemble approach is adding useful information on the uncertainty and spread. Figure 5 shows that the MAE is smaller than the ensemble range for temperature and precipitation in the mid-century and the end of the century for SSP1–2.6. We calculate the ratio of the MAE (centre column) and CMIP6 IQR (left column) and show this in the right-hand column of Fig. 5 to reiterate this point. The range in error of precipitation predictions is higher and more heterogeneous than temperature, although the spatial patterns are similar across timescales. The other scenarios and variables are shown in the Supplement plots (Figs. S10 to S17). The tropics in particular are regions of higher MAE, which is reflective of the differences in the underlying model patterns for these areas, although the MAE is still seen to be smaller than that of CMIP6. Table 1 shows the mid-century and end-of-century RMSE values for each input variable for JULES. RMSE is a standard measure of the error in the predicted variable relative to the mean change. From a previous analysis of the JULES input variables, it is known that there are some variables that are more important for JULES. For example, temperature, specific humidity and precipitation are key drivers with other input variables like wind speed, pressure and longwave downwelling radiation having less influence. This means that even though the pattern scaling for some of these variables may have greater error, they are not as important because JULES is known to be less sensitive to these. Overall, the pattern scaling captures the pattern of change well for the key JULES variables, with the training scenario, SSP5–8.5, having the greatest agreement. This is as expected because not only was this used to generate the patterns but also this is the scenario with the strongest climate change signal. The relative error increases in the lower scenarios related to the need to predict a smaller signal (Wells et al., 2023; Kravitz et al., 2017). However, the low RMSEs for these key variables give us confidence to apply the pattern scaling to different scenarios including stabilisation and overshoot pathways. In future work, we would also like to explore the impact of including the patterns for all of the JULES input variables on the outputs from PRIME in a sensitivity analysis, to understand whether the input variables that do not pattern-scale well but are less important for running JULES affect the spread of the results from PRIME.

We also evaluate whether PRIME pattern scaling can also reproduce the range of changes across the CMIP6 ensemble for all JULES input variables. In several figures we compare time series and end-of-century predicted changes across CMIP6 ESMs for all variables and four regions. Temperature and precipitation are shown in Figs. 6 and 7 with the other six variables in Figs. S18 to S23. The multi-model mean pattern, per degrees Celsius of warming, is shown in the central map for temperature (Fig. 6) and precipitation (Fig. 7). For each region and variable, the figures show time series of change for that region as a shaded plume of CMIP6 output (blue) and predicted by pattern scaling (pink). The end-of-century values for each CMIP6 model individually are shown as a scatterplot for each variable, region and scenario to illustrate the agreement between pattern-scaled and CMIP6 values for temperature and precipitation (see Figs. 6 and 7 respectively) with the end-of-century values also given, along with the RMSE in the regional tables (Table 2 for temperature and Table 3 for precipitation). The Pearson correlation coefficient is a widely used measure of the strength of the linear relationship between two variables; we use it here to quantify the linearity of the pattern predictions against CMIP6 for each model with respect to the one-to-one line. For temperature (Fig. 6) the range of future projections across CMIP6 models spans approximately 5–10 °C warming by 2100 under SSP5–8.5 for each region, with high latitudes warming more than the tropics, as expected. The PRIME pattern-scaled ensemble does well to reproduce this range of projections across the CMIP6 ensemble, with points lying close to the one-to-one line for all regions and scenarios. Pearson correlation coefficients for between-model predictions exceed 0.93 for all regions and scenarios (Table 2), with SSP5–8.5 fitted the best. This is expected as the patterns were derived from this scenario. Importantly, this gives confidence that the PRIME system does not introduce any significant errors, particularly in the training scenario, and that the pattern scaling accurately reproduces the spread of results model by model of the CMIP6 ensemble for this scenario.

Results for precipitation (Fig. 7) also show good agreement, but some mismatches appear as precipitation is more variable in space and time than temperature, as seen in the higher-error characteristics in the pattern scaling and slightly lower correlation coefficients in Table 3. Nevertheless, PRIME predicts well the signal of increasing precipitation over the United States, Siberia and India and reduced rainfall over the Amazon basin. Again, the range and spread of results across the CMIP6 ensemble are well matched, and the correlation coefficients shown in Table 3 are reasonable and above 0.75 for all regions and scenarios.

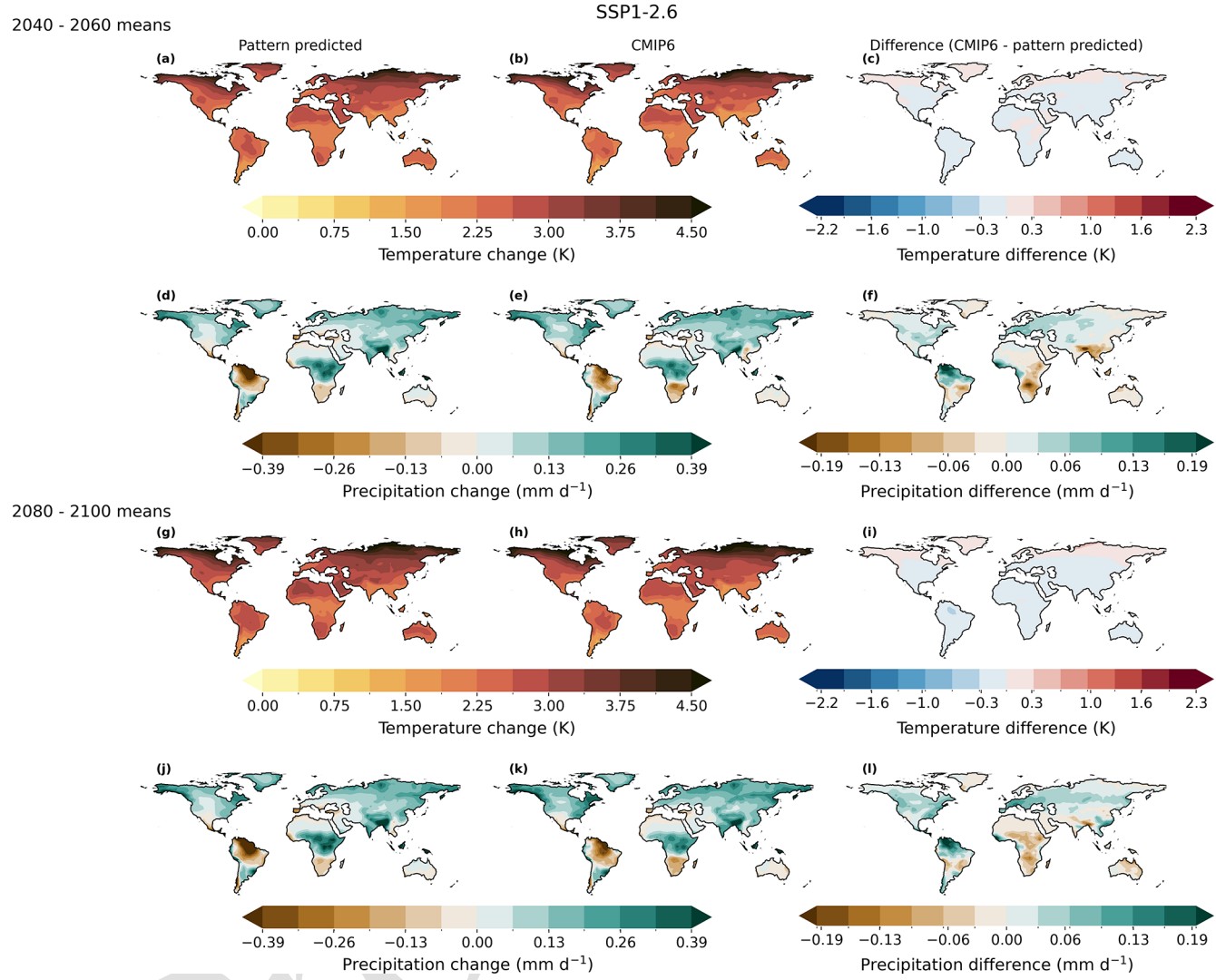

**Figure 4.** Evaluation of the pattern-predicted ensemble mean anomalies compared to the CMIP6 ensemble mean anomalies for near-surface air temperature **(a–c, g–i)** and precipitation **(d–f, j–l)** for SSP1–2.6. Maps **(a–f)** highlight mid-century predictions, and **(g)–(l)** show those for the end of the century. The right-hand column **(c, f, i, l)** shows the difference between the predictions **(a, d, g, j)** and CMIP6 **(b, e, h, k)** the anomalies, in order to show the detail in the prediction error, which is small compared to the change induced by the scenario.

**Table 2.** Root mean square error (RMSE in °C) and Pearson correlation coefficient (Pearson) between pattern-predicted and CMIP6 end-of-century temperature change (see Fig. 6 for scatterplots for each region showing each model) for each scenario. Average values over the region of interest compared to its CMIP6 equivalent by model.

| Region | SSP1–2.6 | | SSP5–3.4-OS | | SSP5–8.5 | |
|---|---|---|---|---|---|---|
| | RMSE | Pearson | RMSE | Pearson | RMSE | Pearson |
| Amazon | 0.27 | 0.96 | 0.22 | 0.98 | 0.13 | 0.99 |
| Siberia | 0.33 | 0.93 | 0.28 | 0.98 | 0.23 | 0.99 |
| USA | 0.21 | 0.95 | 0.21 | 0.98 | 0.21 | 0.99 |
| India | 0.29 | 0.93 | 0.21 | 0.97 | 0.37 | 0.98 |

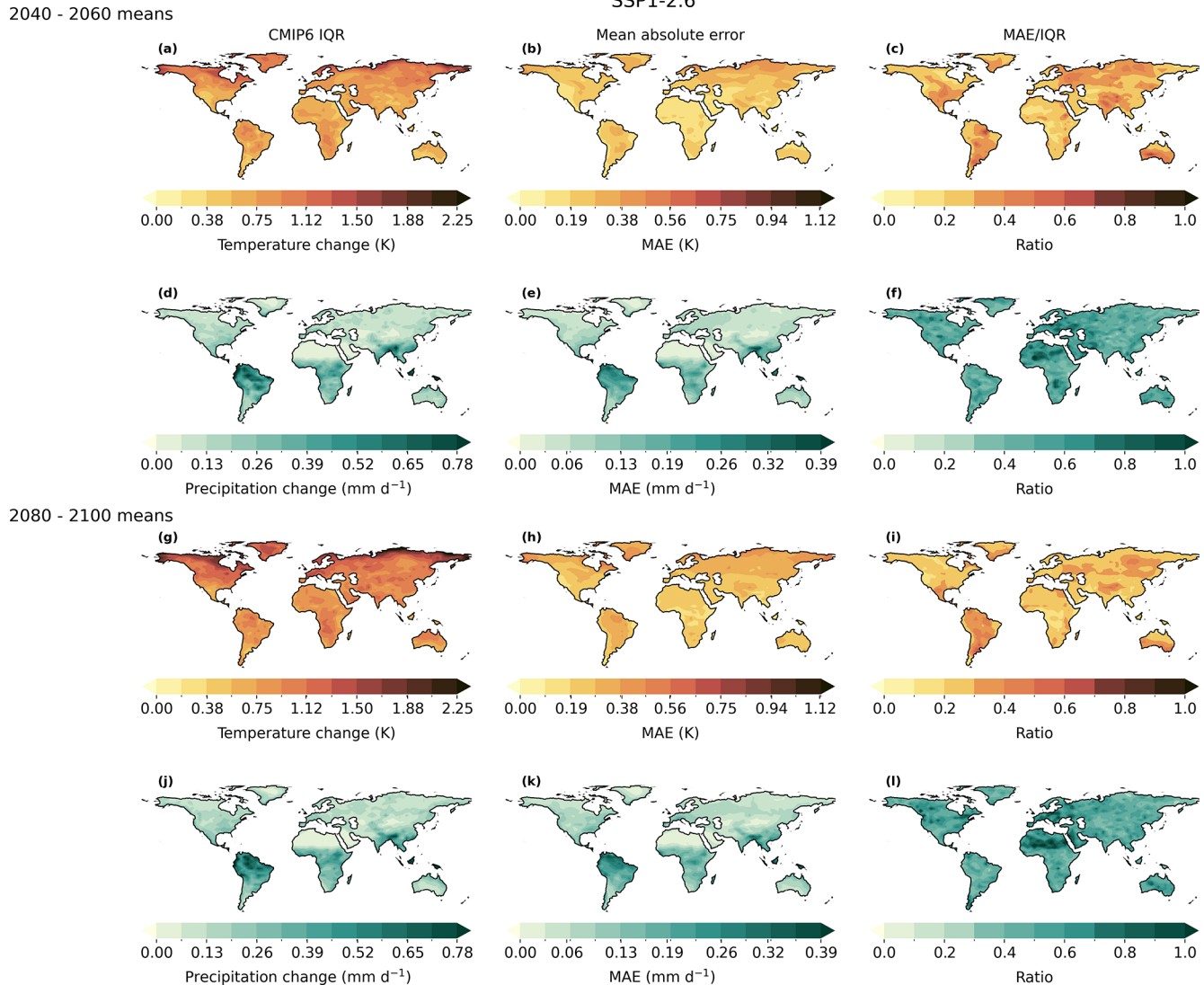

**Figure 5.** Evaluation of the interquartile range (IQR) of predictions **(a, d, g, j)** and of the mean absolute model-to-model error (MAE) for SSP1–2.6 for temperature **(a–c, g–i)** and precipitation **(d–f, j–l)**. Maps **(a–f)** highlight mid-century predictions, and **(g)–(l)** show those for the end of the century. The middle column **(b, e, h, k)** shows the MAE and the right-hand column **(c, f, i,l)** the ratio of MAE to IQR.

**Table 3.** Root mean square error (RMSE in $mm\,d^{-1}$) and Pearson correlation coefficient (Pearson) between pattern-predicted and CMIP6 end-of-century precipitation change (see Fig. 7 for scatterplots for each region showing each model) for each scenario. Average values over the region of interest compared to its CMIP6 equivalent by model.

| Region | SSP1–2.6 | | | SSP5–3.4-OS | | SSP5–8.5 | |
|---|---|---|---|---|---|---|---|
| | RMSE | Pearson | | RMSE | Pearson | RMSE | Pearson |
| Amazon | 0.12 | 0.77 | 0.11 | | 0.85 | 0.079 | 0.98 |
| Siberia | 0.045 | 0.86 | | 0.036 | 0.96 | 0.022 | 0.99 |
| USA | 0.065 | 0.75 | | 0.056 | 0.83 | 0.037 | 0.96 |
| India | 0.094 | 0.87 | | 0.081 | 0.93 | 0.13 | 0.94 |

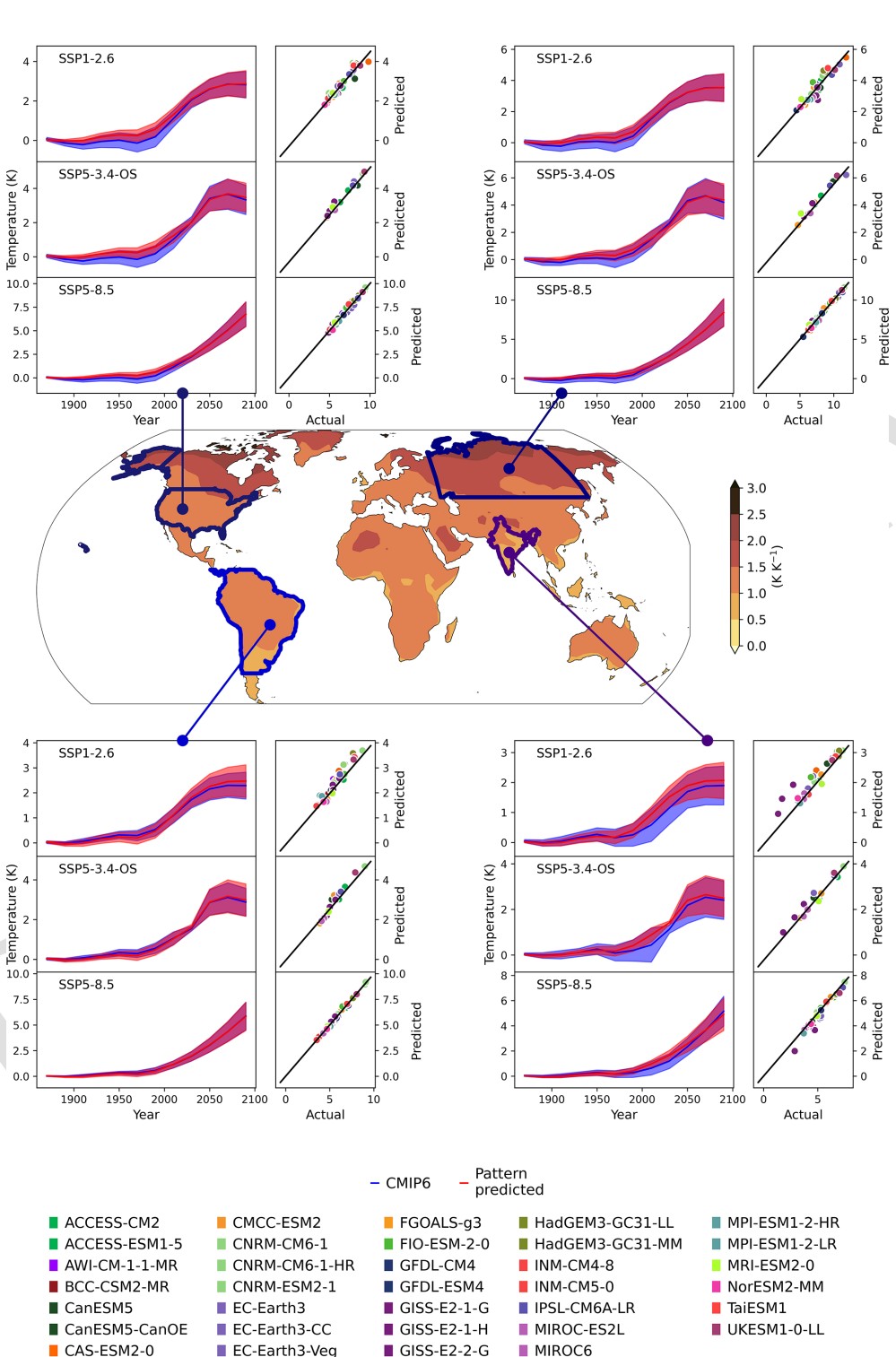

**Figure 6.** The central map shows the temperature pattern (where there is no hatching indicates that the models tend to agree on the sign of the change and with hatching to show where the models tend to disagree on the sign of the change) and subpanels for each region: North America, Siberia, South America and South Asia. The region subpanels show the temperature time series (left subpanel) and scatterplots (right subpanel) for each scenario: SSP1–2.6 (top), SSP5–3.4-OS (middle) and SSP5–8.5 (bottom) (the training scenario). The time series shows the PRIME patterns (blue plume) and the CMIP6 patterns (red plume). The scatterplots show the end-of-century values predicted by PRIME vs. CMIP6 actual values for each model with the model colours shown at the bottom of the figure.

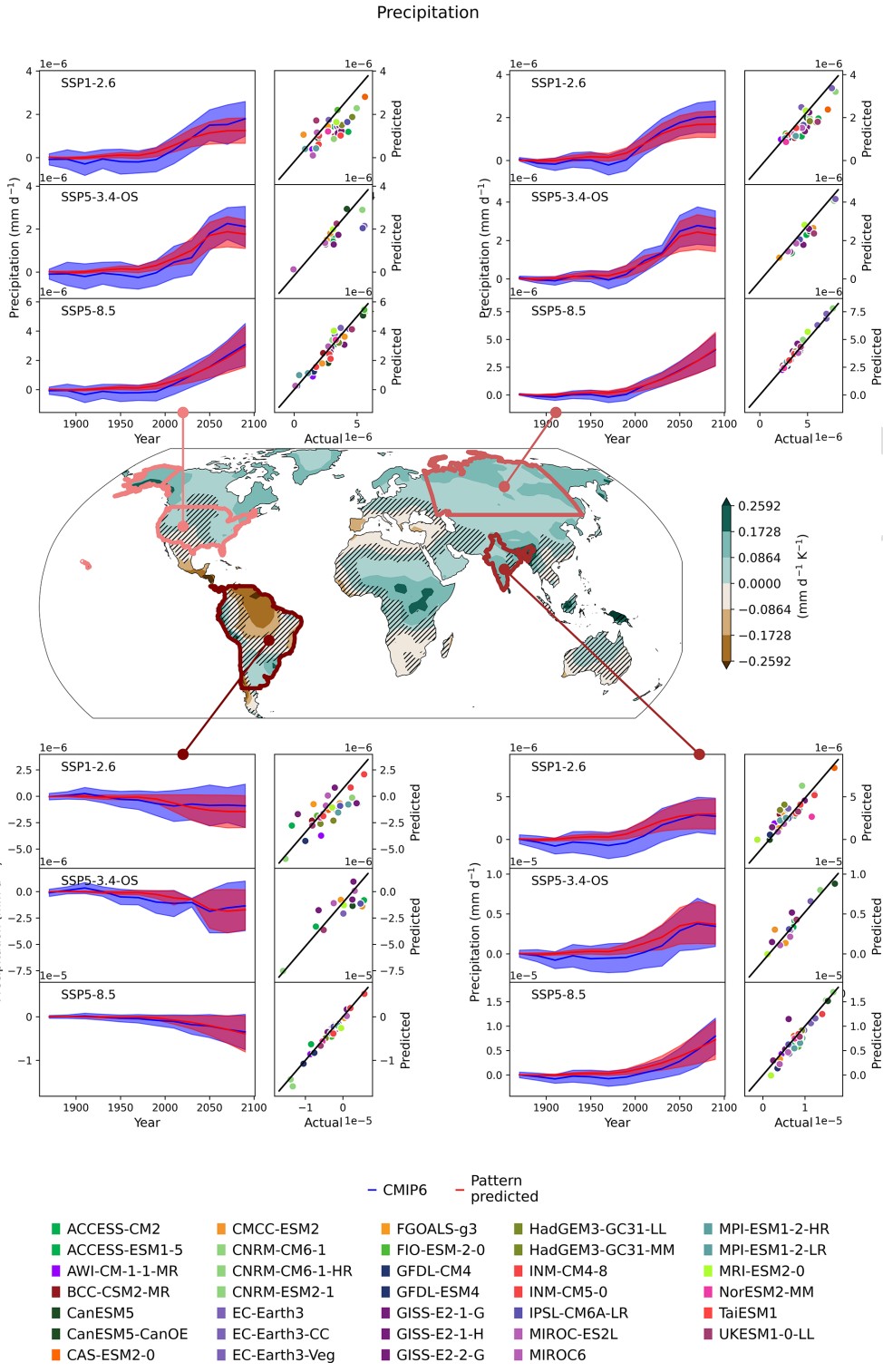

**Figure 7.** The central map shows the precipitation pattern (where there is no hatching indicates that the models tend to agree on the sign of the change) and subpanels for each region: North America, Siberia, South America and South Asia. The region subpanels show the precipitation time series (left subpanel) and scatterplots (right subpanel) for each scenario: SSP1–2.6 (top), SSP5–3.4-OS (middle) and SSP5–8.5 (bottom) (the training scenario). The time series shows the PRIME patterns (blue plume) and the CMIP6 patterns (red plume). The scatterplots show the end-of-century values predicted by PRIME vs. CMIP6 actual values for each model with the model colours shown at the bottom of the figure.

Across-model spread of the other variables (Figs. S18 to S23) is also well captured by PRIME pattern scaling. Changes in humidity (Fig. S18) are well reproduced, while wind speed changes (Fig. S19) have mixed skill, being poorly captured over United States, despite changes in surface pressure (Fig. S20) being well reproduced for all regions. The most notable departure of predicted and actual changes occurs for surface downwelling shortwave radiation (i.e. incoming solar radiation shown in Fig. S21). For all regions and scenarios, the end-of-century values match well, but the significant dip in shortwave radiation during the historical period is not seen at all in the predicted patterns. This period of "global dimming" (Wang et al., 2022; Stanhill and Cohen, 2001) is well known to be caused by anthropogenic aerosols and cannot be replicated by scaling global temperature. Features such as this are an obvious limitation of a pattern-scaling approach, which does not account for different regional patterns from different climate forcers such as aerosols. Finally, changes in downwelling long-wave radiation (Fig. S22) and diurnal temperature range (Fig. S23) are well captured across regions and scenarios by the PRIME pattern scaling.

In conclusion, the patterns both for each CMIP6 ESM and the range of changes across ESMs are generally well reproduced by the PRIME pattern-scaling technique. This is true for each of the four distinct regions and three very different emissions scenarios. The pattern-scaling technique is simple and well understood, and here we find it largely capable of spatially downscaling the global climate response in out-of-sample low-signal and overshoot scenarios.

### 3.3 Land surface and impact simulation

The final section of the PRIME evaluation shows the results using the FaIR-produced projections of global mean surface temperature together with the scaled climate patterns, to drive the JULES land surface model. The JULES step of PRIME is evaluated using two climate variables as examples of output produced by most ESMs. End-of-century changes projected by PRIME are compared against the equivalent ensemble mean CMIP6 data. The example variables considered help us to assess the carbon and hydrological cycles: gross primary productivity (GPP), which is the gross rate of accumulation of carbon via photosynthesis, and runoff, which is the excess water not absorbed by soils and accumulated by water sources.

For this stage of evaluation, we do not expect as good a match with CMIP6 outputs as obtained for the driving climate variables in Sect. 3.2. This is because PRIME uses one land surface model – JULES – which will differ from the embedded land schemes across the range of different CMIP6 ESMs. We perform this comparison for two example variables to demonstrate the extent to which the PRIME framework can reproduce the range of simulated land behaviour from CMIP6, but we can not expect a perfect match. Future

work to include other land models or perturbed parameter ensembles of JULES would help address potential mismatches.

Figure 8 shows the multi-model mean projected end-of-century changes in GPP and runoff in the SSP1–2.6 scenario, the first of two verification scenarios, using both the PRIME framework and CMIP6. Figures S24 and S25 show the equivalent results for scenarios SSP5–3.4-OS (a second verification scenario) and SSP5–8.5 (the training scenario) respectively. The similarity in the predicted spatial patterns can be seen, where in the majority of regions, PRIME matches the pattern of change projected by CMIP6. As we did for climate patterns, we evaluate within and across CMIP6 ESMs. Table 4 presents the mean and interquartile range for both the PRIME and CMIP6 ensemble for each output variable considered. It is important to check that the use of a single land model here does not overly restrict the output and negate the benefits of being able to sample climate sensitivity and climate patterns fully, so while we would not expect PRIME values to be identical to CMIP, we check that the use of a single land model does not result in too narrow a range of outcomes. We see a similar spread and mean for both GPP and runoff (see Table 4). Some deviations are seen between the projections; for example, PRIME projects greater magnitudes of change in both runoff and GPP in the tropical regions compared to CMIP6 (Fig. 8). To put these changes into context from a carbon perspective, PRIME exhibits an end-of-century global increase in GPP in SSP1–2.6 of 26 (between 18 and 34) $\mathrm{Gt\,C\,yr^{-1}}$, while CMIP6 increases by 30 (between 15 and 43) $\mathrm{Gt\,C\,yr^{-1}}$, compared to pre-industrial times. For the training scenario, SSP5–8.5, for PRIME the end-of-century increase is 77 (between 58 and 98) $\mathrm{Gt\,C\,yr^{-1}}$, while CMIP6 increases by 70 (between 37 and 99) $\mathrm{Gt\,C\,yr^{-1}}$, compared to pre-industrial times. Therefore, in both the out-of-sample scenario and the training scenario, PRIME broadly captures the range shown by the CMIP6 ensemble.

Across CMIP6 models, projections are compared in the four specific regions (Amazon, Siberia, USA and India) for both variables (Figs. S26–S29 for SSP1–2.6 and Figs. S30–S33 for the training scenario, SSP5–8.5). PRIME-simulated GPP and runoff can be compared on a model-by-model basis. The results shown in figures in the Supplement (Figs. S26–S33) for each region show CMIP6 output for each ESM and the corresponding PRIME-simulated output from JULES using the climate patterns from the same ESM. For GPP (top row in Figs. S26–S33), the PRIME-simulated changes are typically simulated well, although JULES has a tendency to simulate greater increases in GPP than many of the CMIP6 models. A couple of CMIP6 ESMs clearly stand out. The green lines showing the CNRM model (Séférian et al., 2019) and the cyan lines showing the MPI model (Mauritsen et al., 2019) in CMIP6 consistently simulate greater increases in GPP than JULES. This does not signal an error in PRIME, just that the different land models simulate different sensitivity to future climate changes. However, PRIME does mainly

2080-2100 means, SSP1-2.6

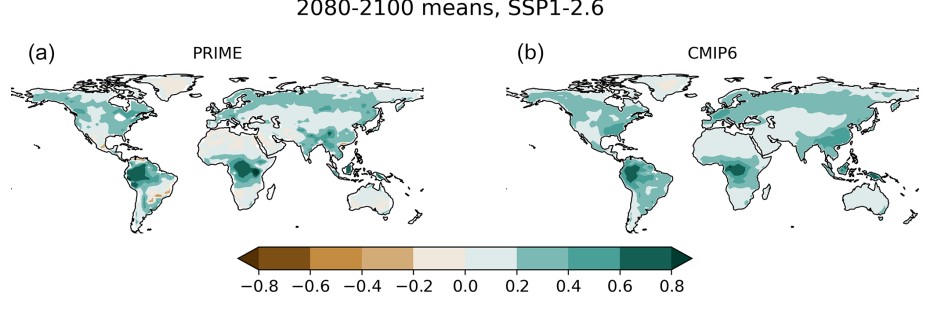

Change in gross primary productivity of biomass expressed as carbon (kg m⁻² yr⁻¹)

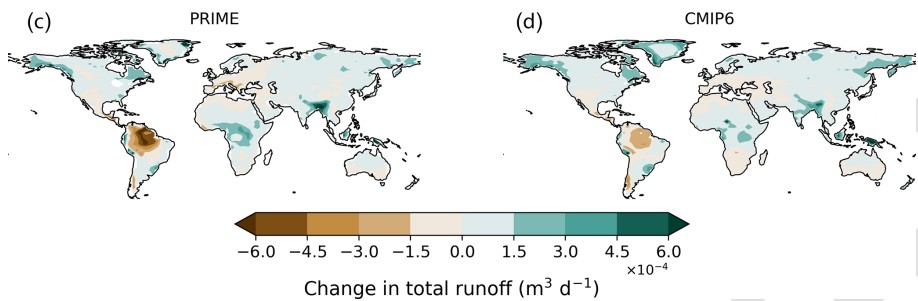

Change in total runoff (m³ d⁻¹)

**Figure 8.** Maps comparing the multi-model mean projected end-of-century changes (2080–2100) for SSP1-2.6 in GPP **(a, b)** and runoff **(c, d)** from PRIME **(a, c)** compared to CMIP6 **(b, d)**.

**Table 4.** Summary table for JULES outputs: mean and interquartile range (IQR) for CMIP6 and PRIME for end-of-century values on land across the globe.

| Variable | Units | SSP1–2.6 | | SSP5–3.4-OS | | SSP5–8.5 | |
|---|---|---|---|---|---|---|---|
| JULES outputs | | CMIP mean (IQR) | PRIME mean (IQR) | CMIP mean (IQR) | PRIME mean (IQR) | CMIP mean (IQR) | PRIME mean (IQR) |
| Gross primary productivity | kg m⁻² yr⁻¹ | 0.22 (0.32) | 0.18 (0.33) | 0.27 (0.43) | 0.24 (0.41) | 0.52 (0.71) | 0.54 (0.87) |
| Runoff | m³ d⁻¹ × 10⁻⁴ | 0.59 (1.22) | 0.55 (1.00) | 0.82 (1.52) | 0.78 (1.32) | 1.61 (2.73) | 2.17 (2.94) |

reproduce the signal and spread of GPP for all regions and scenarios simulated by CMIP6.

Runoff for three of the four regions is well reproduced in PRIME (bottom row in Figs. S26–S33), where Siberia (Figs. S27 and S31), United States (Figs. S29 and S33) and India (Figs. S28 and S32) all see steady increases in runoff consistent with increases in precipitation in those regions. JULES output agrees with these changes of simulated magnitude and spread. The Amazon basin region (Figs. S26 and S30) though exhibits some notable differences. Figure 7 (bottom left) shows a range of precipitation responses over the Amazon with an overall consensus of a drying signal (see also Lee et al., 2021). The JULES outputs though, whilst spanning a similar range of reduced runoff, also show an increase in runoff when forced with some ESM patterns to an extent not shown by the CMIP6 models themselves. The rea-

son for this is not known, but we note that in this case, future projections of Amazon runoff in PRIME show a wider spread than the CMIP6 ensemble. The comparisons shown here illustrate that the PRIME framework gives a good indication of the CMIP6 ensemble spread for these known and very different scenarios to the training scenario. We show a range of different futures including overshoot and mitigation scenarios. This gives us some confidence that we can use this PRIME framework to provide a first look and assess some impacts from scenarios for which ESM simulations do not exist.

## 4   PRIME impact outputs

In this section, we present examples of how the PRIME framework can be used to assess climate impacts. Even though the SSP scenarios have been simulated by many ESMs in CMIP6, only a subset of them simulate the terrestrial carbon cycle (Arora et al., 2020), and very few simulate interactive dynamic vegetation (Pugh et al., 2018). Hence, it is novel to show the possible spread of simulated carbon balance (represented by net ecosystem productivity, NEP) and changes in tree fraction from a sample of percentiles that explore the full range of global temperature sensitivity.

In response to SSP1–2.6 (Fig. 9) and SSP5–8.5 (not shown), terrestrial carbon storage increases almost everywhere in the multi-model mean with positive NEP (top row) especially evident in forested areas. The higher $CO_2$ concentration in the atmosphere drives enhanced vegetation photosynthesis (GPP; Fig. 8), which increases more than any loss from accelerated decomposition. This outweighs any detriment to vegetation productivity from changes in climate except in a few small regions such as southern Brazil. There is, though, significant spread across members with most regions showing potentially positive and negative NEP changes by 2100. This highlights the need for a probabilistic sampling of uncertainty not possible from a limited number of carbon cycle ESMs. We note that this configuration of JULES does not include representation of fire, which has been shown to improve GPP and vegetation distribution in ISIMIP2b simulations (Mathison et al., 2023). In addition, this configuration does not include permafrost carbon dynamics, which could substantially alter this result, as thawing of frozen ground in the high latitudes is expected to mobilise large amounts of organic carbon (Chadburn et al., 2017; Burke et al., 2018; Varney et al., 2023). Both fire and permafrost carbon dynamics are part of planned future JULES configurations to be implemented in UKESM and therefore will be part of future versions of PRIME.

Accordingly, tree fraction increases in all regions (Fig. 9, bottom row). This is robust for India, Siberia and United States with a relatively small spread compared to the mean signal of increased tree cover. In the Amazon region some ensemble members see a stabilisation and even beginning of loss of tree cover by 2100 as the effects of severe climate change counter the benefits due to elevated $CO_2$.

Jones et al. (2023) assessed CMIP6 carbon cycle projections against present-day observations and also saw increases in biomass and total terrestrial carbon storage in all regions throughout the 21st century for SSP3–7.0. That study could not assess changes in vegetation cover as so few CMIP6 models represent dynamic vegetation. PRIME allows us to go beyond CMIP6 results to analyse impacts on vegetation dynamics and ecosystem composition as well as carbon balance.

## 5   Discussion, limitations and conclusions

In this study, we document and evaluate the PRIME framework for the first time, thereby providing a capability for rapid probabilistic regional impact assessments for any global emissions scenario to be produced in a fraction of the time it takes to run an ESM, being able to run hundreds of simulations in just a few days. We have shown that PRIME reproduces CMIP6 results for a range of SSP scenarios that have been simulated by full-complexity ESMs and in doing so demonstrated that the PRIME framework is fit for purpose.

The PRIME framework allows different sources of uncertainty to be quantified. FaIR provides a constrained probabilistic ensemble capturing the uncertainty in climate response. This is informed by the best-available science from the IPCC and can be easily refined or varied to sample any given range of global sensitivity. The advantages of an emulator like FaIR are its efficient run time and ability to provide projections for any emissions scenario outside of those run by ESMs. FaIR is very flexible and can be readily configured to run multiple scenarios, to use multiple parameter sets or to simulate idealised profiles as well as realistic scenarios and pathways.

The uncertainty from the full CMIP6 range of simulated patterns is provided through the construction of spatial patterns of change. It is widely accepted that the spatial patterns of change of many climate variables approximately scale with global temperature and are less dependent on a particular scenario or pathway (Mitchell, 2003; James et al., 2017; Tebaldi and Knutti, 2018; Arnell et al., 2019). As such, it is therefore possible to construct future projections of the spatial pattern of climate change given a pathway of global temperature change. This technique of "pattern scaling", when calibrated against a wide range of climate models, enables a rapid assessment of the range of climates for a given trajectory of global temperature. The limitations of the pattern-scaling method and the potential for developing it are discussed in Sect. 5.1.

In this study we select our ensemble members mainly based on global temperature. However, it is known that rising $CO_2$ concentration also has a direct effect on tropical circulation and precipitation patterns (Bony et al., 2013; Mitchell et al., 2016), which could affect the results shown here. In particular, Mitchell et al. (2016) demonstrate that even if the global temperature stabilises, there is a continuing impact on the precipitation distribution. The joint distributions of the FaIR temperature and $CO_2$ concentrations being used in PRIME are shown in Sect. 3. This illustrates that although we capture the full temperature range through our selection of ensemble members based on global temperature, the higher $CO_2$ concentrations are not as well sampled. This is a limitation of the method we have chosen to select the ensemble members rather than a limitation of the PRIME framework,

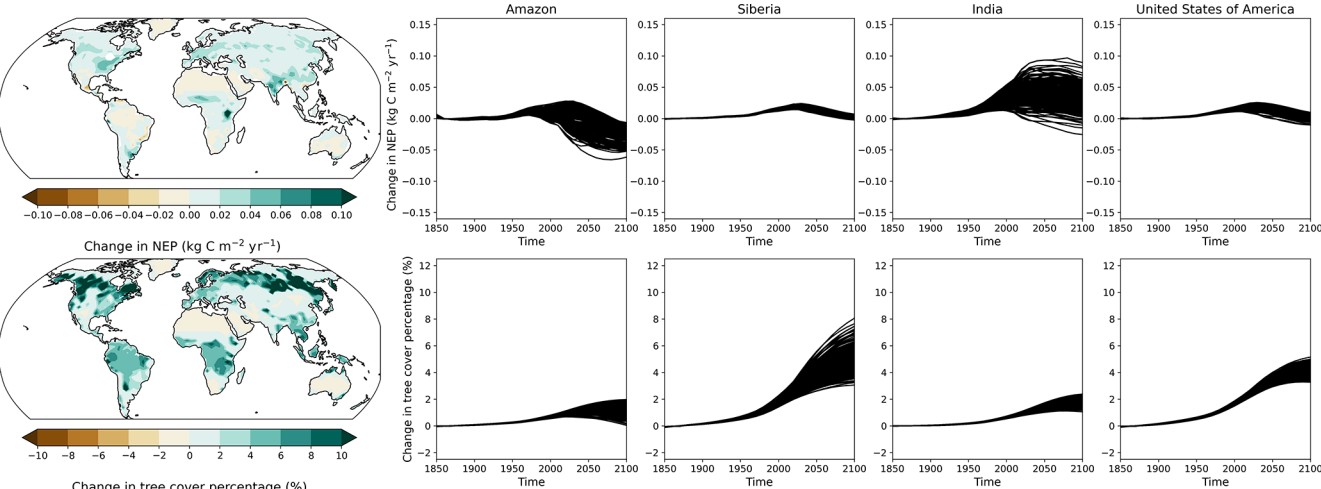

**Figure 9.** Maps of net ecosystem production (top panels) and tree fraction (bottom panels) with time series showing PRIME output for each ensemble member for each study region: Amazon, Siberia, India and the USA (labelled) for SSP1–2.6 between 1850–2100.

and this will be explored in future iterations of the framework.

One of the main advantages of the PRIME framework is its flexibility, with the simple coupling between components lending itself to future couplings using other downstream models. For example, adding other models such as one for sea level rise or air quality to this framework would expand the scenario and climate output to get a rapid response of a broader range of impacts beyond land. However, this simple one-way coupling between components, while a benefit in terms of flexibility, could also be deemed a limitation because changes in emissions from the land are not allowed to feed back on the scenario. This is a desirable capability that we plan to build into PRIME, but in its current form, the structure of the framework does not allow this to operate. The comparison with CMIP6 outputs shown here draws on simulations which are "concentration-driven". Therefore, this feedback onto atmospheric $CO_2$ is not included even for carbon cycle ESMs, which makes the analysis shown here, comparing CMIP6 simulations and PRIME, a clean comparison. Currently, we include just one land model, but other land models could be included in addition to JULES to capture the structural uncertainty in land models as well. It is also worth noting that the methods for downscaling to subdaily timescales in the form of the weather generator in JULES could benefit from more modern approaches which have not yet been investigated herein.

### 5.1  Pattern-scaling limitations and opportunity for development

Pattern scaling is a powerful methodology which has grown from its original intended purpose of providing a technique to allow extension of the relatively few computationally intensive simulations by ESMs to a broader remit. However,

there are two main limitations of this methodology. First, by definition, pattern scaling assumes the local and monthly changes in climate to be linear in global warming. Yet many studies show non-linearities in the climate response (King, 2019; Osborn et al., 2018; Chadwick and Good, 2013) and that the climate system may contain "tipping points" where strong non-linearity implies there may be future times when there are strong responses of Earth system components to relatively small additional increases in greenhouse gases (e.g. McKay et al., 2022). Linear scaling will not capture such rapid changes if they impact near-surface meteorology, although investigation of ESMs reveals relatively few instances of rapid change (Drijfhout et al., 2015). Although the use of pattern scaling can currently only offer a linear interpretation of local and seasonal near-surface meteorological response to increasing greenhouse gases, the inclusion in PRIME of the full JULES land surface model does offer the opportunity to investigate in detail the risk of tipping points in land ecosystem response because JULES includes aspects of plant physiology and vegetation dynamics that are strongly non-linear. A second limitation of pattern scaling is that it does not resolve local land–atmosphere feedbacks and so will not capture in full the effects of major alterations to the land surface on near-surface meteorology. Such feedbacks may occur by the addition of new processes to land simulations that adjust substantially land–atmospheric exchanges of sensible or latent heat flux. Development of techniques to include such local feedbacks will form the basis of future research. Additionally, pattern scaling assumes that the patterns do not change with time, and studies such as King et al. (2020) have shown changing spatial patterns of climate on long timescales as the system begins to equilibrate following an initial transient period. Yet experience with the MESMER tool has shown only marginal improvement when additional

predictors of patterns are added to global temperature, with the simple, conventional pattern-scaling approach showing significant skill with errors typically much smaller than intermodel spread (Beusch et al., 2022).

There are alternate approaches to pattern scaling such as the "time-shift" approach (Herger et al., 2015; Schleussner et al., 2013; King et al., 2017), which assumes that scenarios with equivalent global mean temperatures exhibit similar regional climate changes. For example, a time period from an early transient high-forcing simulation could be used to represent a climate sample for a lower-forcing scenario. The advantage time shifting offers is that it avoids the linearity assumption and maintains physical consistency across multiple variables. However, there are parts of the climate system that are influenced by climate forcing rather than with global mean temperature. For example, Ceppi et al. (2018) show a poleward shift of the mid-latitude jets and Hadley cell edge in response to changes in forcing even before half the warming response has been realised. Furthermore, the history of the climate forcing as well as the balance of different forcing agents (which may evolve differently across different scenarios) also influences the regional climate change in scenarios with the same global mean temperature response.

Neither the approach of traditional pattern scaling nor "time-shifting" is without limitations, with both providing useful capability. However, there is a need for fuller evaluation of pattern-scaling approaches, including aspects such as wind or snow cover, where shifts may not scale with global mean temperature. An intercomparison of models like PRIME and MESMER would be a valuable addition to the literature. In addition, it would be useful to explore the use of multiple predictors such as land–sea contrast for more slowly evolving processes, along with $CO_2$ and aerosols for their direct effects. Ongoing research into land use and direct regional biophysical effects will also be brought into subsequent versions of PRIME. For the future, PRIME is well positioned to exploit rapidly developing artificial intelligence (AI) and machine learning (ML) methods, for example, Mansfield et al. (2023), Kitsios et al. (2023), Wilson Kemsley et al. (2024) and Mansfield et al. (2020) to name a few. These offer substantial advances in deriving downscaled and interpolated data, which will be an area of development for PRIME.

## 5.2 Conclusions

Overall we have shown that PRIME is a flexible framework that runs quickly and produces reliable results for known scenarios. PRIME reproduces the climate response to a range of emissions scenarios (within the known limitations of the pattern-scaling approach) spanning global temperature in close agreement with IPCC assessments, capturing a range of 34 state-of-the-art Earth system models, and simulating a range of land surface outcomes and impacts. Although there are some variables that do not pattern-scale as well as temperature, the performance of the key JULES input variables represents the range of CMIP6 models. This gives us confidence that PRIME will enable rapid and probabilistic assessment of novel scenarios, thereby providing useful insights and the capability to quantify societally relevant climate impacts.

*Code availability.* FaIR v1.6.2 is available from the Python Package Index at https://doi.org/10.5281/zenodo.1247898 TS4 (Smith et al., 2022), on GitHub at https://github.com/OMS-NetZero/FAIR/tree/v1.6.2 (last access: December 2024) and on Zenodo at https://doi.org/10.5281/zenodo.4465032 Smith et al. (2021). Climate pattern calculation code is available from ESMValTool https://doi.org/10.5281/zenodo.12654299 TS5 (Andela et al., 20124).

*Data availability.* FaIR output used in PRIME is available from Zenodo at https://doi.org/10.5281/zenodo.10524338 TS6 (Mathison and Smith, 2024).

The ESMValTool pattern recipe linked above automatically downloads the CMIP6 data from ESGF https://doi.org/10.5281/zenodo.12654299 TS7 (Andela et al., 20124) and calculates the patterns.

JULES output for the variables shown for each scenario is available from Zenodo at https://doi.org/10.5281/zenodo.10634291 (Burke and Mathieson, 2017).

Calibration data for FaIR v1.6.2 are available from https://doi.org/10.5281/zenodo.6601980 (Smith, 2022).

*Supplement.* The Supplement is provided in a separate PDF file called "Supplementary_for_PRIME_doc_paper.pdf". The supplement related to this article is available online at [the link will be implemented upon publication].

*Author contributions.* CM, EJB, CJ and CH came up with the original concept implemented here and contributed to running some of the individual components of the framework. GM created the patterns; EK completed the analysis of the individual components of the framework, which make up many of the plots. CS provided expert knowledge on FaIR, and CJ and AJW contributed scientific ex-

pertise particularly around the carbon cycle and JULES. LKG and LJV contributed to discussions and writing the manuscript. All were involved in bringing the ideas together and writing and commenting on the manuscript. DM contributed expertise at the revision stage on statistics and evaluation.

*Competing interests.* The contact author has declared that none of the authors has any competing interests.

ther geographical representation in this paper. While Copernicus Publications makes every effort to include appropriate place names, the final responsibility lies with the authors.

*Acknowledgements.* The authors would CE2 like to thank Tim Andrews for providing his perspective on the manuscript. Thanks also go to the three reviewers and the editor for their contribution to this paper.

*Financial support.* This work was supported by the Joint UK BEIS–Defra Met Office Hadley Centre Climate Programme (GA01101), the Newton Fund through the Met Office Climate Science for Service Partnership Brazil (CSSP Brazil), the Natural Environment Research Council (NE/T009381/1), and the European Union's Horizon 2020 research and innovation programme under grant agreement no. 101003536 (ESM2025 – Earth System Models for the Future). Norman J. Steinert CE3 was funded by the Research Council of Norway (project IMPOSE, grant no. 294930) and the Norwegian Research Centre AS (NORCE). Chris Huntingford received support under national capability funding as part of the Natural Environment Research Council UK-SCAPE programme (award no. NE/R016429/1). Rebecca M. Varney was supported by the European Research Council's Climate–Carbon Interactions in the Current Century project (4C; grant no. 821003). Chris J. Smith was supported by a NERC–IIASA collaborative research fellowship (grant no. NE/T009381/1) and the European Union's Horizon Europe research and innovation programme under grant agreement no. 101081661 (WorldTrans).

*Review statement.* This paper was edited by Jinkyu Hong and reviewed by three anonymous referees.

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

**Remarks from the language copy-editor**

CE1    Please give an explanation of why this ("Amazon" to "South America" throughout) needs to be changed. We have to ask the handling editor for approval. Thanks.

CE2    Please note that "also" was removed here. The second sentence was also moved from the financial support.

CE3    Please note the slight adjustments to this section, in line with house standards.

**Remarks from the typesetter**

TS1    I would like to inform you that I cannot download your new Supplement because a login and password are required. Please send me a link where I can download the Supplement without having to create an account. I kindly ask you to remove the title page information in your Supplement because it will be added automatically during the publication process. Thank you in advance.

TS2    Please check all changes concerning the authors and their affiliations very carefully.

TS3    Please confirm changes.

TS4    Please confirm change to DOI number.

TS5    Please confirm change to DOI number.

TS6    Please confirm change of DOI number.

TS7    Please confirm change to DOI number.

TS8    Please provide all author names.

TS9    Please provide all author names.

TS10    Please provide all author names.

TS11    Please confirm addition.