# Peer review of "A rapid application emissions-to-impacts tool for scenario assessment: Probabilistic Regional Impacts from Model patterns and Emissions (PRIME)"

_EGUsphere, 2023_

## Author Comment (AC1)

**Response to reviewers**

Thank you to both reviewers for your detailed review of our manuscript. We have responded to each comment in full and outlined the changes we will make to the manuscript to address your comments in this document. Our responses are in black font in response to review comments in blue, and where we quote new text, this is in italic.

**Response to reviewer 1:**

Reviewer comments in blue and the response in black below.

The authors propose a global downscaling system based on "pattern scaling" where the global-mean temperature is used to predict local mean values of all variables on a GCM grid, which is then used to drive an offline land model in order to calculate climate impacts. This approach is considerably cheaper than running a GCM and is proposed for generating large ensembles and testing more complete sets of scenarios while still providing regionally specific outputs.

**Author's response:** Thank you for acknowledging this, it is why we are thinking along these lines.

I think this system is worth publishing but have some issues to raise that will require moderate revisions.

**Author's response:** We aim to address these point by point below.

General Comments

1. The success of the pattern scaling approach is not well tested in my opinion, mainly because most of the tests examine the change by end of the century under the RCP8.5 scenario, which is exactly the same one used to determine the pattern; and the change by end of century will dominate the variance that a max-likelihood linear fit is trying hardest to fit. Thus the success is built in by design and the agreement shown in e.g. Fig. 5 is meaningless. What is needed is an out-of-sample test such as the accuracy at mid-century, and/or for other RCP scenarios—the RCP6 or 4.5 scenarios would seem like the obvious test targets. The time series comparisons (Figs. 6,7) look OK but not great, as there are some errors mid-century that are as large as the signal. This suggests that the pattern scaling approach isn't as accurate as we'd like.

**Author's response:** We only show SSPs in this paper (there are no RCPs used) although we show SSP5-8.5 in the main text, this is for illustration of how we have conducted the analysis. SSP1.2-6 and SSP5.3-4-OS are shown in the supplementary data.

In short, the paper needs to get rid of RCP8.5 tests and instead show tests on at least two other RCPs to give a realistic idea of out-of-sample performance.

**Author's response:** We do 2 things here - we show SSP5-8.5 results to show that PRIME does not introduce any errors and could reproduce the SSP5-8.5 scenario used in calibration; this is the minimum we would hope and expect for this scenario. Additionally we test the system on 2 very different scenarios - Currently the figures for the SSP1-2.6 and SSP5-3.4-OS analysis are in the supplementary data, both of these scenarios are very different to SSP5-8.5, with one being an overshoot scenario. We also include values covering the analysis for each of these scenarios in the tables in the main text. This demonstrates that we broadly capture the CMIP distribution for these scenarios. To address this comment and make this clearer, we will make the signposting to the analysis for these other scenarios clearer in the manuscript and point to the values in the table. We will also move the SSP5-8.5 figures to supplementary and use SSP1-2.6 in the main text.

2. One reason the performance isn't always good, especially on precipitation, may be that the authors are ignoring the direct effects of CO2 on precipitation which are substantial (e.g. Bony et al. 2013). Past studies show that by combining the effects of CO2 and global-mean temperature, precipitation patterns can be well captured, but not based on temperature alone. Since the authors are already feeding CO2 and global-mean T to JULES, why not also use CO2 as a second predictor for the downscaling?

**Author's response:** Thanks for the comment, the CO2 response is, of course, implicit in the calibration because it is part of the simulations from which we create the patterns. We discuss limitations of pattern scaling in the discussion and will extend this to bring out future research priorities. Other predictors such as CO2 but also land-sea contrast will be considered. We will add greater depth to our discussion of limitations and development priorities in the discussion section (this also addresses similar comments from reviewer 2). Specifically in the revised manuscript, we will include discussion of the effect of rising atmospheric carbon dioxide concentrations on tropical circulation and precipitation referring to the Bony et al., (2013) paper and mention the Mitchell et al. (2016) paper that highlights that temperature is not the only indicator for precipitation distribution, which is also affected by non $CO_2$ forcings (Mitchell et al., 2016). Mitchell et al. (2016) also show that there is not an absolute 1:1 link between global temperature and global precipitation, indicating that even if the temperature stabilises there is a continuing impact on precipitation distribution. This is also something else we would like to explore with this framework in future iterations.

3. I am very confused by the so-called "weather generator" since the text states that the same weather is used for every day of any given month (line 128). If so, that is extremely unrealistic and will produce extreme responses in the land model (since it will either rain every day of the month or not at all). This doesn't sound like an actual weather generator. Little else is said about the weather generator except to cite Williams and Clark 2014—we should not have to look there to get basic information about what kind of weather is being inserted. If indeed the weather is being held constant for a whole month at a time and then switches to something else on the 1st of the next month, this needs to be highlighted as a significant limitation in discussing the results.

**Authors response:** In this framework we use the weather generator that is built into IMOGEN to downscale to the hourly timescale and accurately simulate the diurnal cycle and exchange of heat, water and momentum and to avoid numerical instabilities, this is similar to the disaggregator described in Mathison et al. (2023). The IMOGEN weather generator distributes monthly mean

rainfall subject to a probability distribution that has fixed parameters in time (i.e. year), although dependent on month and location. For each year, a random number generator is applied to sample from the distribution. The distribution parameters are fitted to known historical gridded measurements of precipitation. Precipitation is considered to occur in a single event, with a globally specified duration parameter (6 h for convective rainfall, 1 h for large-scale rainfall and convective snowfall and large-scale snowfall). Convective rainfall occurs when temperature exceeds 288.15 K. Sub-daily downward shortwave radiation and temperature are estimated using known factors such as the position of the sun in the sky and a sinusoidal function to represent the maximum and minimum daily range. Downward longwave radiation is a linear function of temperature, and specific humidity is kept below saturation at each time step. We will add text to the manuscript to say this more clearly.

4. The probabilistic framework being used is not clear from the description. Any probabilities will depend on the priors for example, which are not stated, and on what observations the probabilities are conditioned on which is also not stated. There are also some confusing statements in the text (see detailed comments below). This needs to be clarified if the intention is for this tool to be used for probability estimation. It looks like the pdfs are traced to an ensemble calculated by WGI of AR6 but still the assumptions should be stated here.

**Author's response:** In this paper we are presenting a framework – PRIME – which can, and will, be used in multiple ways. Individual users have to choose their own sampling of all the components from FaIR percentiles to CMIP patterns. There is nothing intrinsic in PRIME which mandates this choice.

Secondly though we will better explain what we have done here – as an example use. We have sampled within the FaIR temperature distribution (but of course we could use other percentiles and/or $CO_2$ levels). But this is not in itself a limitation of the system. These are not intended to be particular priors in nature and the output is not presented in a probabilistic way.

PRIME can be sub-sampled to reduce the number of simulations and also to create probabilistic output. It is not an intrinsic part of the tool though, and PRIME could be run with all possible combinations of parameters and patterns. As such this documentation paper does not recommend any specific sampling strategy or choice of prior. This is for individual users of PRIME to determine on a case by case basis.

In this paper, we use the AR6 FaIR calibration (fair-calibrate V1.4.0, Smith et al.,2024) which takes into account the observed historical temperatures up to 2015, at which point the SSPs begin. The prior distributions come from a 1.6 million member prior ensemble. This large ensemble is reduced using the historical temperature record eliminating those members with a larger error, with the aim of reproducing the uncertainty range in present day relative to Pre-industrial. We then simultaneously constrain on several observable and assessed climate metrics including ECS, TCR, Aerosol, $CO_2$ concentration and ocean heat content change. This reduces the ensemble to approximately 2237 members, as used in AR6. We take a selection of ensemble members from a single scenario that are designed to span the range of the temperature change distribution from FaIR and we use these ensemble members throughout to allow comparison between scenarios.

We do not currently use the $CO_2$ concentration to select the ensembles but we do use it as an input to JULES. We will make this process clearer in the text.

**Detailed/Technical Comments**

line 183: by constraining future projections here, do you mean constraining equilibrium climate sensitivity?  This is not the same as constraining RCP projections directly (which depend on factors other than ECS, most importantly historical forcings).

**Author's response:** This sentence is correct as it stands. AR6 constrained future projections (of global temperature) refers to the process that was used in AR6. This process included a much broader set of constraints and was not only based on the ECS. See IPCC Chapter 4 (Lee et al), Section 4.3.4. In our calibration of FaIR, however, ECS is not a constraint but an emergent property we include as just one of several factors to define the distribution; the need to use other metrics aside from ECS to describe climate uncertainty was made in Smith et al. (2023) [Climate uncertainty impacts on optimal mitigation pathways and social cost of carbon - IOPscience]. A more detailed response to this question is given to point 4 above.

line 196: this is stated a bit confusingly—I assume the CO2 is a forcer to the land model, not to the meteorology (which depends only on global-mean temperature).

**Author's response:** CO2 is a forcing for the land model in addition to the meteorology. We will clarify this in the text, in which the word "secondary" is confusing.

line 200-202: I don't understand why in your ensemble, CO2 and ECS would be correlated. I think this is because the important elements noted in Major Point 4 are missing.  Even if you are conditioning on historical warming I don't see why a higher future CO2 would imply a lower ECS. This would only make sense if you were targeting a specific warming, but that isn't stated clearly and you are showing a spread of possible warmings for any given RCP, as occurs in standard GCM simulations where a prior is placed (implicitly and usually independently) on both ECS and carbon cycle parameters and this implies a posterior distribution of temperature at any future time. There are a number of past studies that obtain pdfs of future warming conditional on historical warming using EMICs, and this study should follow a similar approach;  most of them use the GCM ECS distribution as a prior but some use observationally-constrained priors on ECS.

**Author's response:** As noted above, we will clarify that PRIME can be used to sample anywhere within the distribution of FaIR outputs and climate patterns and used for any scenario. As a framework there is no constraint on the assumptions or priors chosen. The optimal sampling strategy can and will vary on a case-by-case basis depending on the desired use of the tool. We will explain better in the text the choices we have made, but also stress that these choices are not intrinsic to PRIME as a tool.

Fig. 3: y axis or caption needs to identify at what time the CO2 concentration is determined.

**Author's response:** We will make it clearer that the CO2 concentration and the temperatures relative to 1850-1900 are both defined at 2100.

Fig. 4: upper right figure panel needs to specify what humidity it is (specific humidity, according to the text).

**Author's response:**  We will modify the label to say specific humidity

line 220-222: I think what this text means is that you are correlating the decadal means of the predicted vs. CMIP variables—please state clearly. I would not say the correlations are very good for precipitation, wind etc., since much of the map is around .4 or less which means only 20% of the variance is captured by the emulator.

**Author's response:**

We will clarify these lines in the text, which do describe correlating the decadal means of predicted variables against CMIP variables.

**Response to reviewer 2:**

Reviewer comments in blue and the response in black below.

**General Description:**

The authors present a combination of three existing model approaches – a global climate model emulator (FaIR), a traditional pattern scaling approach, and the JULES land model. They term this chain of models PRIME, suggesting that "PRIME correctly represents the climate response for [these] known scenarios,..." and that "PRIME enables the state-of-the-art science to be used throughout the modelling chain starting from the latest scenarios all the way to the simulations of regional impacts."

**Author's response:**  Thanks for this comment. In general when we talk about PRIME we use the term Framework more often than Tool because it does indeed comprise a series of components run in sequence. We propose keeping the title as it is, but amend the language in the paper to explain that a stand alone tool is the general ambition and direction of travel for this framework. A high priority technical development is to set up a PRIME rose suite that will allow the framework to be run in an automated way. However, it is possible to reproduce the analysis presented here with the data and model information we are providing as part of this submission.

**Overall Comment:**

If the paper were presented with a heading like "Pattern-scaling approaches to drive JULES" or something similar, then I think it would be a great addition to the scientific literature – as it is a fine example of how the chain from global emission scenarios to some land-based impact metrics can be made with a number of (simplified) assumptions. However, the paper presents itself as playing in a different league, e.g., to "bypass ESMs" (line 66), or implicitly suggests that ISIMIP bias-corrected ESM outputs could be replaced (lines 25 to 40ff). With these heightened expectations, I'm sorry to say that the paper is underwhelming.

**Author's response:** Thanks for drawing our attention to this, we are disappointed by this comment, but I think this was more as a result of our enthusiasm for the framework and project than intentionally over stating the capability. However, to address this we will look to tone down the language we use and expand on the limitations of the approach currently mentioned in the discussion (Line 362-376). This modification to the text may mean that we add a section on the limitations and development opportunities for this framework during the editing of the manuscript.

We respond to the individual reasons below:

The reasons are:

- The chosen method to justify the adequateness of pattern scaling (e.g., Fig 4): The authors show the Pearson correlation coefficients between scaled patterns (derived apparently from a linear regression of CMIP6 output against global mean temperature) and the CMIP6 data. I am very confused about this choice, i.e., to use a Pearson correlation coefficient for "Evaluation" (see e.g., caption of Figure 4). Suppose there is no change in regional precipitation in a specific region under climate change. If the pattern scaling "gets it correct" and indicates zero changes for those grid points, the applied Pearson correlation coefficients would be around zero (as there is no linear relationship then between pattern scaling and CMIP6). Thus, authors should use either standard RMSE (see Chapter 3 of IPCC AR6 WG1) or other useful metrics – or explicitly justify their use of the Pearson correlation coefficients.

**Author's response:** We respond to this comment below as this comment also relates to the next comment.

- Unclear p-values and low skill for 5 out of 8 variables: In Figure S2, the authors show the percentage of p-values, averaged over models and months. First, I couldn't find any statistical description of what null hypothesis was tested. That there is no climate change? The authors are strongly urged to complement the paper with a detailed statistical section that both illuminates their suggested uses of Pearson correlation coefficients (or ideally other evaluation metrics) and the p-values here in Figure S2. On the substance of it: Only three out of the eight variables are shown to have satisfactory 'p-values' of <0.05 (whatever exactly was measured here). How can the authors claim the low percentages of <0.05 p-values (take e.g., Northern Europe or North America precipitation changes) as an indicator that "PRIME correctly represents the climate response for these known scenarios" (Abstract, line 16f.). That seems a really far-fetched conclusion given the presented results in Figure S2 and Figure 4, which seem to suggest that for precipitation, the PRIME pattern scaling results are essentially useless for many key world regions.

**Author's response:** Here we respond to both of the previous reviewer 2 comments on the evaluation of the patterns.

We agree that our evaluation of the pattern scaling using Pearson correlation coefficients could be improved through the use of alternative metrics. On this basis, we will find more appropriate and descriptive metrics for the evaluation, and ensure that we clearly distinguish our analysis of pattern scaling performance on other scenarios versus the one the patterns were fitted against (SSP5-8.5).

Figure 4 and S2 were intended to summarise the general performance of temperature as a predictor for each of our meteorological variables across all the model patterns, but we accept that in trying to condense the analysis into one figure, this is confusing and is therefore the opposite of

the useful summary it was supposed to be. Therefore, we will replace these figures with ones that break down the pattern scaling evaluation in more detail, with clearer metrics and language in the analysis.

We will present analysis to demonstrate the pattern scaling's ability to capture the variance and mean of the CMIP6 ensemble for each scenario, variable, and for each month on which the patterns were trained. We will highlight more clearly the variables for which the patterns do well and also those where they do less well and explain why we still use them. In particular, we know from previous analysis of the input variables for JULES that there are some variables that are more important for JULES. For example, temperature, precipitation and specific humidity are key drivers with other input variables like wind speed, pressure and longwave downwelling radiation having less influence. The revised evaluation of the patterns may take the form of an example plot using (RMSE and mean response for the multimodal ensemble for a given month) with others available in a Zenodo repository. We will also revisit the maps that show the difference between the predictions and CMIP6 values at the end of the century and the data in the tables to ensure their meaning is more transparent. In future work, we would also like to explore how including patterns for all of the JULES input variables affects the outputs from PRIME in a sensitivity analysis.

- Almost the same as a study from a quarter-century ago. As the authors state, PRIME is very similar to the year 2000 pattern scaling approach by Huntingford and Cox. Indeed, the two papers have almost a very similar scope, using the TRIFFID model instead of JULES. And arguably, the study from almost half a century ago uses more elaborate timeseries plots and statistics to showcase the merit (despite the general strong limitations of pattern scaling for the majority of variables).

**Author's response:** Thank you for the comment, the reviewer is obviously aware of the literature of pattern scaling over many years and will know that the current authors – one of whom led the first IMOGEN study - have been central to the development and use of IMOGEN in multiple studies since the initial Huntingford and Cox (2000). PRIME represents a logical evolution of the IMOGEN pattern scaling tool – each component can be optimised and developed, and we agree with the reviewer that such a tool is a very valuable addition to the research community.

In response to this comment, we will more clearly present the history of this framework and its connection with the previous work of Huntingford and Cox (2000). We will explain more clearly in the text why PRIME is a new development that builds on the work of Huntingford and Cox (2000). These will include highlighting the following:

- Specifically, in this current paper the energy balance approach of IMOGEN is now replaced by FaIR which enables it to draw on a wide research community of reduced complexity modellers, and pick up developments to FaIR from across this community. The current use in PRIME enables direct linkages to the extensive work which calibrated FaIR for use in IPCC AR6. This is a substantial strength over IMOGEN. FaIR incorporates the latest science from AR6, including a broad range of gas species which influence the global temperature. For example we are able to incorporate representation of short lived forcers like methane and short-lived halogenated compounds, such as hydrofluorocarbons (HFCs), hydrochlorofluorocarbons (HCFCs), nitrogen oxides (NOx), carbon monoxide (CO), non-methane volatile organic compounds (NMVOCs), sulphur dioxide ($SO_2$), and ammonia ($NH_3$). This means we are not restricted to only including the influence of $CO_2$.

The pattern scaling approach is inspired by and based largely on the code in IMOGEN, which is very much the same as Huntingford and Cox (2000), but linking this existing modelling capability with JULES and FaIR in this way is a first and substantially increases the range of variables that we can consider using from this framework.

- The pattern scaling technique for now is the same as used previously, but now updated to use CMIP6 models. Previously this laborious step was carried out manually. We now automate this and make the ESMValtool recipe freely available.
- The patterns themselves of course have evolved significantly from "a quarter of a century ago" and now represent the latest state-of-the-art ESMs from CMIP6. As a framework these can now be easily updated with future CMIP generations too.
- JULES has evolved substantially from the land model used in that study too, now representing much improved plant physiology, hydrology, vegetation types and river flow (Mathison et al.,2023 ISIMIP paper). PRIME as a framework enables rapid use of the latest science in JULES to be pulled through as and when further developments are implemented, such as fire and permafrost.

Overall we are unashamedly excited by the potential of this tool, and do not see its heritage back to Huntingford and Cox as a weakness.

- Science has evolved since 2000. Probably most fundamentally, I am concerned about the following point. For many of these five variables, the literature is far more progressed and established that simple pattern scaling does not work satisfactorily. It works like a charm for regional mean temperature, even extremes to some extent, but not for precipitation, wind, pressure, etc.
    - Take for example precipitation. As Allen and Ingram pointed out in 2002 (https://doi.org/10.1038/nature01092), or Andrews et al., 2010 (https://doi.org/10.1029/2010GL043991), the hydrological cycle underlies various constraints, but is not only driven by global or regional surface air temperature changes. The vertical changes in the troposphere's energy budget, i.e., the GHG radiative forcing itself as well as the aerosol cloud interactions, have a substantial effect on precipitation.
    - Take for example wind and storm tracks: The key feature is that midlatitude storm tracks might move poleward due to a broadening of the Hadley cells (https://doi.org/10.1038/s41561-017-0001-8). That is fundamentally at odds with simple linear pattern scaling, which scales the response at one location just up.

**Author's response:** Thank you for this comment.  In the IPCC AR6 (sec 4.2.4 on pattern scaling) they acknowledge that precipitation can be pattern scaled – the slow response of precip is forcing-independent, and the fast response is not significant for all forcings (e.g. solar), but issues (esp. with aerosols) do indeed complicate things. As such AR6 used the epoch/time-shift approach. This is a possible avenue for future generations of PRIME, but for now we continue to use the established pattern scaling approach. Our results show that precip patterns and the variations across individual CMIP models are well captured on a regional basis for all 3 scenarios, especially in the latter half of the century. Figure 7 has detailed regional, time profile and model-by-model results which are summarised in the statistics in Table 3. We believe this is compelling evidence that PRIME captures general trends and importantly the spread between models for future precipitation changes in different world regions under very different scenarios.

To address this in the manuscript we will include a section on limitations of PRIME, including pattern scaling, and future development priorities which include:

- A call for fuller evaluation of pattern scaling approaches – including aspects such as winds, or snow cover where shifts may not scale with global T. An intercomparison of models like PRIME and MESMER would be a valuable addition to the literature.
- Use of multiple predictors such as land-sea contrast for slow evolving processes, and maybe CO2, aerosols for their direct effects. Ongoing research into land-use and direct regional biophysical effects will also be brought into PRIME.
- Alternative techniques to pattern scaling including epoch/timeshift (Herger et al. (2015), Schleussner et al., 2016; King et al., 2017) as adopted by IPCC for GWLs.
- AI and ML techniques offer big advances in deriving down-scaled and interpolated data. For both pattern scaling and weather generation we expect substantial advances and PRIME is well positioned to exploit them.

- The somewhat poor alignment of PRIMAP results with observations for the historical period underscores these fundamental shortcomings of pattern scaling when venturing outside temperature, specific humidity, and longwave downwelling radiation (which is not too different from lower tropospheric air temperature). Similarly to Huntingford and Cox (2000), Zelaszowski et al. (2018), and others, the skill of regional precipitation patterns remains very low to the extent of not being very useful.

  **Author's response:** While we agree that precipitation does not pattern scale as well as temperature, Figure 7 shows that for some regions the predictions are within the range of CMIP6 models especially when looking at end of century values. See response above to comments about pattern scaling of precipitation in AR6.

- PRIME is not open source, as the underlying JULES code is not open source, if I understand correctly.

  **Author's response:** JULES is going through a process to become open source so I hope to be able to update this with more information and reassure this reviewer that PRIME will shortly be fully open source.

- PRIME does not seem to be a tool in itself. In the code availability section, unless I overlooked it, there is no PRIME code available. The pointers are to the underlying energy balance model, to the ESMVALtool (which has tons of functionality beyond patterns), and to JULES (upon request). Thus, the paper seems to describe a sequence of how to apply three other models in sequence. Yet, PRIME is described as a 'tool' itself. What am I misunderstanding?

  **Author's response:** See response to comment on the general description above, the aim is for this to be a framework that can be easily rerun to reproduce the results shown here. We are working toward this as a priority but it does not preclude publication of the current PRIME framework and results.

**Overall Recommendation:**

If the authors undertake major revisions, possibly a combination of back-scaling the bold expectations that they raise among readers, with additional strong skill statistics and an extensive set of limitations, I think the paper can be a very valuable contribution. That is because systems like the proposed PRIME one are definitely needed. The demand for probabilistic climate impact projection systems is definitely there. However, the paper has to be honest about the various shortcomings, rather than claiming that it correctly represents the climate projections of CMIP6 models (e.g., line 16 and similar at many other places). The system that Huntingford and Cox (2000) described was not too dissimilar, to be frank.

**Author's response**: Thank you for this comment, please see response above to similar sentiment in the general comments.

**Detailed Comments:**

- Line 15: 'which was used to define patterns'. A stricter delineation into "training" and "verification" data throughout the manuscript would be appreciated. For example, it is not clear whether Figure 4 is based on the SSP5-8.5 data or not. It better not be.

  **Author's response:** While we train on SSP5-8.5 and use this scenario mainly to show that there is no odd behaviour introduced by the framework, we specifically test on scenarios that are very different in SSP1-2.6 and SSP5.3-4-OS. We will therefore remove from the manuscript any figures that show SSP5-8.5 and put these figures in the supplementary information, replacing them with figures that show SSP1-2.6.

  Overall, we will revise the manuscript to ensure figures/results are appropriately described drawing a clearer distinction where we are describing the patterns versus where they are being evaluated.

- Line 19: 'being used for impact assessments'. The monthly average patterns are maybe sufficient for some impact patterns, but a simple pattern scaling approach that, for example, completely loses the covariance between temperature and precipitation extremes is not useful for all impact studies. As mentioned above, a bit more precise wording would be appropriate (or a bit more humbleness, or both).

  **Author's response:** As mentioned above, we will go through the manuscript and update the language as appropriate.

- Line 47 ff.: A more comprehensive discussion of the many similarities and few differences to the rich history of pattern scaling approaches seems useful. Certainly the 1999 Huntingford & Cox study, the Mitchell review paper, etc.

  **Author's response:** As stated above, we will include a section in the resubmission to recap on the lineage of the PRIME framework. As some of the same authors of those studies, we are well aware that this represents an evolution of ours, and others', previous work. We do not pretend to have

invented the subject here. Some of the history of IMOGEN will be included either in the manuscript or supplementary information using these references:

- Combining the GCM analogue model with the JULES land surface model to create a full land impacts system. That's : DOI10.5194/gmd-3-679-2010
- A first attempt at fitting to the non-UK ESMs was with CMIP3, here: DOI10.5194/gmd-11-541-2018
- Pattern scaling from Mitchell et al. (2001)

- Line 66: 'Bypassing ESMs' is strong wording – and I would say inappropriate. At best, the proposed approach can approximate a few key characteristic outcomes of ESMs.

  **Author's response:** We will look to rephrase this to explain that this framework is intended as a first look at the scenarios where ESM simulations have not yet been run or will never be run because the ESMs require so much more in terms of resources (including setting up and inputs). Therefore this framework could actually inform which scenarios the ESM simulations should or could run.

- Line 73: 'PRIME enables the state-of-the-art-science throughout the modelling chain'. It would be good if the authors can explain that a bit better, specifically with regards to changes in variability, compound risks, etc. (Or choose less hyperbolic wording).

  **Author's response**: We will rephrase this to explain that PRIME facilitates faster pull-through of state-of-the-art science to propagate from the latest scenarios and regional climate change patterns (from the latest ESMs) to the simulation of regional impacts.

- Line 109ff: So, if I understand the method correctly, only 9 runs are undertaken with the energy balance model – scanning the stated percentiles (BTW: It would be interesting to hear how high the author's confidence is in the min-max values, the 0% and 100% percentile). In general, I think the methods section needs to be much expanded, and the authors should clarify how exactly these 9 EBM runs are combined with various patterns? Also, how does the methodology cater for hot and dry futures versus hot and wet futures? Is any covariance preserved across the variables?

  **Author's response:** In PRIME we have the full ensemble of FaIR runs from AR6 which consists of 2237 ensemble members. PRIME as a framework is not constrained to any choice of prior or sampling, but in the interest of keeping the PRIME ensemble to a manageable size for this paper we select 9 of these that are representative of the temperature distribution and span the range of uncertainty in temperature from FaIR.

  As outlined in Section 2.2, we apply the output from FaIR as a scaling factor to each of the patterns from 34 CMIP models. We generate the patterns separately from the CMIP models using the code provided in ESMValTool. We then use the IMOGEN code within JULES to generate 3-hourly data with which to run JULES. This means that PRIME is run for *each* CMIP pattern individually (we do not run it using the average CMIP pattern). And so explicitly it considers all combinations of GCM output for hot/wet and hot-dry combinations. We will make this clearer in the text.

- Line 115 et al: As mentioned above, the methodology description does in its current form not allow reproducing the study. I would argue it should – even though the code is allegedly provided. For

example, over which time window was the regression undertaken? Including all points from 1900 to 2100 or just the last twenty years from 2081-2100 in the SSP5-8.5 run?

**Author's response:** See response above in which we state that we intend to generate a PRIME Rose suite that will enable anyone to run PRIME, this is a work in progress. We will add to the methods section that the regression was calculated with points from the duration of SSP5-8.5, from 2015-2100. The patterns calculation from the CMIP6 model data is available to run as an ESMValTool recipe, as cited in the code availability section.

● Line 205, Figure 3: How does this plot show the validity of the chosen approach? Even though the underlying energy balance model is able to produce a joint probability between CO2 concentrations and global mean temperature, the sampling of the 9 percentiles is only sampling a fraction of the distribution in the CO2 concentration dimension?! The sampled CO2 concentrations in PRIME span 950ppm to 1100ppm, while the energy balance model output suggests a range from 800ppm to 1200ppm?! In some regions (such as the Amazon), an 800ppm or 1200ppm CO2 concentration might well produce a different physiological plant response, yet the probabilistic PRIME model does not seem to propagate that information forward?

**Author's response:** As stated above – PRIME can sample any/all – we show here results where sub-sampling was kept to a tractable level. But yes of course any specific study can and should sample appropriately. In this study we focused on the temperature distribution as a first attempt at the approach, but we also felt that we should be transparent and point out that in carrying out the selection with a focus on the temperature distribution, we do not capture the full spread of the CO2 distribution and this should be mentioned.

● Line 230, Figure 5: If SSP5-8.5 data is used to derive the PRIME patterns, it is hardly a useful comparison to show a comparison between SSP5-8.5 CMIP6 and PRIME. That is like showing training data as independent verification, which misleads about the skill of the model.

**Author's response:** We use multiple SSPs to evaluate PRIME. These are shown explicitly and discussed in the manuscript and supplement. Figure 5 will be swapped to one of the other scenarios currently in the supplementary information.

● Line 238 and vicinity: The authors just skim over the fact that when compared to other scenarios that are not used for deriving the patterns, the skill gets worse (partly understandable because of the lower signal-to-noise ratio in lower scenarios). The authors need to unpack that with quantifiable information and more details (i.e., which periods, which models were looked at, what is the difference across CMIP6 models, how is the transient skill before the end of the century, etc.).

**Author's response:** The different skill for other scenarios is quantified in the tables and suppl. Plots. Temperature timeseries for all 3 scenarios and 4 example regions are shown in figure 6 and precip in figure 7. Other variables are in the supplement. Tables 1-3 also show quantitative info across scenarios and variables.

● Line 238: When the authors write: 'However, the high correlations and low RMSE give us confidence to apply the pattern scaling…' I am not sure what 'high correlations' the authors refer to? The SSP5-8.5 ones from which the patterns were derived or the lower SSP scenario ones? If the former, then the high correlations should not give confidence to anyone that the model is skillful

outside its training scope. If the latter, then the detailed plots in the supplementary are providing the usual picture that linear pattern scaling provides: That it is very good for some variables, and absolutely unusable for others. For example, Figure S8 on shortwave downward radiation shows the 'prime' (pardon the pun) example, where pattern scaling with global mean temperature does NOT work.

**Author's response:** Please see our response regarding revision of the evaluation of the patterns noted above.

- Table 1: It would probably be useful to the reader if the RMSE values are put into the context of the mean change of the respective variables.
- Table 1: The Pearson correlation coefficients that are stated seem oddly high and I can't reconcile them with the regionally disaggregated ones shown in Figure 4. For example, take precipitation in Figure 4… judging from the color scale, the global-mean Pearson correlation coefficient (if it were meaningful, see above) would be somewhere between 0.3 and 0.7. Yet, the results for all three SSPs show values well above 0.83 in Table 1, even 0.97 for SSP5-8.5!? Please append with much more methodological detail and/or code of what exactly was done. And please explain, why those are consistent.

**Author's response:** During revision, we will revise the evaluation section. The RMSE values are our principal evaluation metric and we agree the mean change would be useful alongside. We will extend this to all the evaluation tables. We acknowledge some variables do not pattern scale well. However, we find that for our selected 'impact' metrics the framework performs well.

- Line 313: 'These comparisons allow us to confidently use the PRIME framework to assess impacts.. ' – Again, I think this is another example of slightly overconfident language that does not appropriately reflect the various limitations.

**Author's response:** See previous responses explaining that the authors will add more information on the limitations of the method and revise the language where appropriate.

- Line 317: The authors write: "Hence it is novel to show a full probabilistic range of the possible spread of simulated carbon balance (represented by NEP)'. See above. I have my doubts, whether 'full probabilistic' is the right term here.. as e.g., the CO2 concentration uncertainties do not seem to be explored according to Figure 3.

**Author's response:** We will look at rephrasing this to explain that we only properly sample the spread of the temperature distribution.

- Line 359: The MESMER tool showed some improvements for surface air temperature when using additional predictors. That regional surface air temperature is the variable that is stunningly well already predicted with a pattern scaling approach. Nobody challenges the usefulness of pattern scaling for monthly mean regional temperatures. The authors seem to want to suggest here though that this marginal improvement would also be true for pattern scaling more generally. Really? I highly doubt it given the literature on scaling regional precipitation, for example, where GHG forcing, (regional) aerosol forcing have clearly been shown to not only provide marginal improvement but are vital predictors without which precipitation cannot be adequately projected (see above).

**Author's response:** Thanks for this comment, which we agree with. This speaks to PRIME being a framework as opposed to a modelling protocol. A user could replace pattern scaling with a regional climate emulator of their choice. Incidentally, we were pleasantly surprised with the fairly good

performance of pattern scaling for precipitation, and as a first demonstration of the process chain will retain its use in the paper. Of course, other more sophisticated emulators exist (PREMU, MESMER-M-TP, fldgen 2.0) that will probably do better for precipitation. In fact, a rapid impacts model intercomparison project, FastMIP, is in progress that will investigate some of these questions. Providing that the driving climate data for the regional emulator is a combination of outputs of FaIR (or any other simple climate model) - generally global mean surface temperature and the radiative forcing from particular greenhouse gases, aerosols, and natural forcings, then any regional emulator could be used. This still may be insufficient for downwelling shortwave radiation which depends partly on the regional pattern of aerosol forcing.

- Line 384: The authors write "Overall we have shown PRIME faithfully reproduces the climate response… ". If authors would phrase this conclusion to something like "Within the known limits of the linear pattern scaling approaches, we have shown that 3 out of the investigated 8 variables can adequately be projected in their individual monthly means'… or similar, I would have no issue with it. But conclusions like the one above are way overconfident in my view.

**Author's response:** Thank you we will revise the concluding statement to better represent the limitations of the methodology.

- I can see how much dedicated work went into this manuscript, which is why I apologize that I cannot be more positive. With major revisions, I think this manuscript can add a useful contribution to a very important field – but the current form of the manuscript requires an overhaul from multiple angles in my perception.

**Author's response:** Thank you for the detailed review.

References to add to the paper and used in this response:

Smith et al. (2024): https://egusphere.copernicus.org/preprints/2024/egusphere-2024-708/egusphere-2024-708.pdf

Smith et al. (2023): Climate uncertainty impacts on optimal mitigation pathways and social cost of carbon - IOPscience

Mathison et al., 2023: GMD - Description and evaluation of the JULES-ES set-up for ISIMIP2b (copernicus.org)

Huntingford, C., Cox, P. An analogue model to derive additional climate change scenarios from existing GCM simulations. *Climate Dynamics* **16**, 575–586 (2000). https://doi.org/10.1007/s003820000067

Bony et al., 2013: https://www.nature.com/articles/ngeo1799

Mitchell et al., 2016: https://www.nature.com/articles/nclimate3055

Mitchell et al., 2001: https://crudata.uea.ac.uk/cru/pubs/thesis/2001-mitchell/timm.pdf

---

## Referee Report (RR1)

Mathison et al. describe a new climate emulator that is coupled with a land-based impact model, which together form PRIME. This paper describes a significant advance in the use of FaIR, a commonly used 1D climate emulator because it allows users to simulate regional climate using patterns from many different climate models at relatively low computational cost. Furthermore, it allows for rapid impact assessment with its direct integration with an impact model.

I do not have major methodological concerns, but I have several questions and suggestions that I would like the authors to address before recommending publication. The more general ones are written here and several smaller line-by-line suggestions are listed below.

The underlying assumption of matching an ensemble of FaIR output with an ensemble of climate model patterns is that there is no relationship between the two. In other words, the patterns in warm models do not look significantly different from the patterns in cold models. In some FaIR papers, specific climate models are emulated by using tuned parameters (e.g. Leach et al. 2021, Figures 3 and 4), but here the full range of models of global mean T is then matched with a full range of patterns. To address this, I could imagine a supplemental figure that, for example, plots the pattern correlation between each model and the multi-model mean, against the global mean temperature in a future year (e.g. 2050 or 2100). There may be other ways to do this, but I think a small additional analysis to demonstrate that the pattern is not a strong function of sensitivity would be useful to demonstrate whether this is a limitation or a non-issue. I would be surprised if this was a major problem, but I think it would be worth characterizing for potential future users.

As for other users, I'm wondering whether the pattern scaling component is a stand-alone model. The schematic (Figure 1) shows patterns that include the ocean, whereas the rest of the figures are limited to the land. I could imagine many applications in which a user would want an ensemble of patterns-scaled climate (including the ocean) but is not interested in the land-based impacts. It would be helpful if you could address that kind of hypothetical use case in the text.

Lastly, I think the text about the land model needs to acknowledge that atmospheric carbon (and other variables) have no feedback with the changes in vegetation. Although I still see use in this application, I think this deserves more serious consideration in the text, especially as comparisons are made to CMIP6, where some models do simulate the terrestrial carbon cycle (line 398). This seems like a significant difference between PRIME and some GCMs.

My comment on line 164  about adding anomalies from a 1850-1889 baseline to 1901-1930 climatology may also require some changes if indeed I'm understanding the methodology correctly.

Minor:

Line 4: I think "global picture" is a confusing choice of words. Also there are other climate emulators that have a spatial element, so I'm not really sure what this sentence is trying to say.

Line 5: "general information" is vague

~60: the overview of ongoing efforts is great. I think the machine learning models like Climatebench probably deserve a mention for completeness, since they have similar objectives

66-68: I found this sentence confusing. "this type of input"? Not sure what input is being referenced

98-99: Choices affect the way patterns are selected? Do you just mean users choose the patterns? Also, please define Rose suite.

135: As stated repeatedly, part of the appeal of emulators is the low computational cost. However, the sampling done here is fairly sparse (Figure 3) and the reason given here is to "make the ensemble size manageable". I think it would be worth dedicating a bit of text to explain this discrepancy.

136: "these are selected using one scenario so that scenarios can be compared against each other" is confusing

147-148: Do the linear regressions include an intercept? If so, how is it treated?
Please also specify one realization of each model (i.e. not an ensemble and not all available)

152: I think there needs to be a summary of this conversion from monthly to 3-hourly data. Not necessarily here, but somewhere.

159: "this is the method currently used in PRIME" - it's not clear what is being referred to

161: I thought the methods were described quite clearly up until paragraph, where it starts to get confusing. It's somewhat unclear what belongs to IMOGEN and what does not. I also don't understand why the diurnal cycle calculation could cause numerical instabilities (166) when there are no feedbacks. And what part of this is 3 hourly and where is the monthly temperature from the pattern scaling coming into play? Is there only one diurnal cycle for each month?

164: Why are 1850-1889 anomalies added to 1901-1930 climatology? If it's observationally limited (i.e. no data prior to 1900), then it seems like the anomalies should also be relative to 1901-1930? Maybe you could at least demonstrate that this is a reasonable approximation, but it seems inconsistent. Also why did you decide to add it to an observational climatology instead of a modeled one?

Equation 3: I think that 3 should be an exponent - please check the equation

194: typo - mode, not model

213: I don't think you mean "where ESMs have not been run" because the simulations exist

Figure 2 and S1. Caption should specify where the data are from. It says in Fig S1, but not in Fig 2. Could you also mention in the text again why you're subsampling? It hasn't been mentioned in several pages

Figure 4: I don't think the differences should be on the same scale as the anomalies. For example, 4c just shows that most differences are less than 0.6 degrees, but grouping 0.01 to 0.6 together does not make sense when almost all values are less than 0.6. Plus the colors are faint (for the same reason) so they are hard to see.
The figure captions (for 4 and accompanying SI figures) should also specify "anomalies" for the "predicted ensemble means"

Figure 5: I think adding a ratio column (MAE / IQR) would better make the point.

S15: All colors in all of the related figure are shown from light to dark, except pressure in S15. Although it's not wrong, I would recommend reversing the colors for clarity

307: This sentence and several sentences in this paragraph are a little wordy and could be edited for clarity. For example, the reference to Burton et al does not seem to fit well in the context, and "actual values" on line 315 is vague.

345: What does it mean for the pattern scaling technique to be well understood by the literature?

366: I'm a little confused here - you would not expect the values to be identical but "hope for a similar spread". I agree it seems to work impressively well, but there is some disconnect between saying there are several differences (e.g. only using one land model) and yet expecting similar spread. The rest of the paper is about emulating spread when using many models, so why would you expect the spread here to be similar when only using one model? A bit of clarifying text would be helpful.

Figure 9: The caption says "median and uncertainty ranges" but I don't see that in the figure. I am confused about what the lines are - can you clarify in the text and caption? Are they for different pattern scaling from each of the different climate models?

469: You mention that local land-atmosphere feedbacks are not included, but that seems secondary to me given the fact that the land model produces carbon-cycle relevant parameters like GPP that do not feed back into FaIR. As mentioned above, I think that qualification should be added.

---

## Author Response (AR2)

**Response to reviewers**

Thank you to the reviewer for your detailed review of our manuscript. We have responded to each comment in full and outlined the changes we will make to the manuscript to address your comments in this document. Our responses are in black font in response to review comments in blue, and where we quote new text, this is in italic.

**Reviewer general comments**

Mathison et al. describe a new climate emulator that is coupled with a land-based impact model, which together form PRIME. This paper describes a significant advance in the use of FaIR, a commonly used 1D climate emulator because it allows users to simulate regional climate using patterns from many different climate models at relatively low computational cost. Furthermore, it allows for rapid impact assessment with its direct integration with an impact model.
I do not have major methodological concerns, but I have several questions and suggestions that I would like the authors to address before recommending publication. The more general ones are written here and several smaller line-by-line suggestions are listed below.

Thanks for your comments, we respond to each comment in turn here and in the revised manuscript.

The underlying assumption of matching an ensemble of FaIR output with an ensemble of climate model patterns is that there is no relationship between the two. In other words, the patterns in warm models do not look significantly different from the patterns in cold models. In some FaIR papers, specific climate models are emulated by using tuned parameters (e.g. Leach et al. 2021, Figures 3 and 4), but here the full range of models of global mean T is then matched with a full range of patterns. To address this, I could imagine a supplemental figure that, for example, plots the pattern correlation between each model and the multi-model mean, against the global mean temperature in a future year (e.g. 2050 or 2100). There may be other ways to do this, but I think a small additional analysis to demonstrate that the pattern is not a strong function of sensitivity would be useful to demonstrate whether this is a limitation or a non-issue. I would be surprised if this was a major problem, but I think it would be worth characterizing for potential future users.

Thank you for this insightful comment. We note that the literature suggests a link between the strength of feedbacks and the pattern of warming, e.g. Andrews and Webb (2018), Ringer et al. (2014). However, it is the difference in atmospheric physics that dominates the spread in climate model climate sensitivity (Dong et al. 2020) and is what FaIR is aiming to emulate. From this perspective, we expect the pattern effect to be a second order uncertainty. Furthermore, when considering our overall aim of sampling uncertain climate impacts the evidence from CMIP6 is that the a 'hot-model' is not necessarily correlated with a positive bias in the simulated impact (Swaminathan et al., 2024). Swaminathan demonstrates quantitative evidence that for a range of impacts metrics (such as flood, drought and fire weather) there is at most only weak correlation between these impacts and climate sensitivity. In other words there is no pattern dependence of being a "hot" or a "cold" model. In fact Swaminathan et al., (2024) goes further and suggests that to try to assume a link from the patterns to the global temperature can actually be misleading and lead to omission of some important impact-relevant information. The challenge then is to appropriately constrain the patterns without artificially reducing the impact spread. This is an area we will

certainly take forward in future assessments and revisions. In summary, our approach is to sample the combined uncertainty in climate and pattern effects noting that we may be oversampling the spread.

In response to this comment we have therefore added the following text to the manuscript:

*"Our approach combines the full range of FaIR temperature responses with the full range of CMIP ESM patterns. We note a pattern effect relating warming to climate sensitivity (Andrews and Webb (2018), Ringer et al. (2014)) has been shown in the literature. However, assessments of simulated impacts in the CMIP6 ensemble sampling a wide range of impacts metrics from multiple regions found little or no correlation with climate sensitivity for most regions and climate drivers (Swaminathan et al., 2024), which contributes to justifying the approach to treat these independently. Other studies have found changes to circulation patterns and dynamical regimes more important for climate patterns than global scale thermodynamical response (Ribes et al., 2021 and 2022, Palmer et al 2023). To maximise our sampling of uncertainty we therefore take the pragmatic decision to co-vary all patterns with sampled temperature pathway."*

As for other users, I'm wondering whether the pattern scaling component is a stand-alone model. The schematic (Figure 1) shows patterns that include the ocean, whereas the rest of the figures are limited to the land. I could imagine many applications in which a user would want an ensemble of patterns-scaled climate (including the ocean) but is not interested in the land-based impacts. It would be helpful if you could address that kind of hypothetical use case in the text.

Thank you for this comment, although the patterns generated include the ocean we chose to focus our analysis on land because JULES is a land surface model and only outputs data on land. Our focus for this framework has been mainly to look at Earth System and climate impacts on land as output from the JULES model when run as an impacts model. However, part of the benefit of running a framework like this is being able to output at various points along the pipeline, so we have added this possibility by including the following to the text:

*"We generate global patterns that include land and ocean but in this analysis, we focus on the patterns over land for running JULES and considering land impacts. However, it would be possible to use the patterns over the ocean and exclude the land for other downstream applications."*

Lastly, I think the text about the land model needs to acknowledge that atmospheric carbon (and other variables) have no feedback with the changes in vegetation. Although I still see use in this application, I think this deserves more serious consideration in the text, especially as comparisons are made to CMIP6, where some models do simulate the terrestrial carbon cycle (line 398). This seems like a significant difference between PRIME and some GCMs.

My comment on line 164 about adding anomalies from a 1850-1889 baseline to 1901-1930 climatology may also require some changes if indeed I'm understanding the methodology correctly.

We agree this could be an important feedback, we now discuss our aspiration to add it to a future development of PRIME. For now the structure of the framework does not allow this to operate. The comparison with CMIP6 outputs though is still a clean (like-for-like) one because we draw on simulations which are "concentration-driven" and so, even for carbon-cycle ESMs, also do not allow this feedback

onto atmospheric $CO_2$ to operate. We have now acknowledged this with the following text in the discussion:

*"However, this simple one-way coupling between components, while a benefit in terms of flexibility, could also be deemed a limitation because changes in emissions from the land are not allowed to feedback on the scenario.This is a desirable capability that we plan to build into PRIME but in its current form, the structure of the framework does not allow this to operate. The comparison with CMIP6 outputs shown here draws on simulations which are "concentration-driven" therefore this feedback onto atmospheric $CO_2$ is not included even for carbon-cycle ESMs, which makes the analysis shown here, comparing CMIP6 simulations and PRIME, a clean comparison."*

We address the comment on line 164 in the specific comments section below.

**Specific Comments:**

Line 4: I think "global picture" is a confusing choice of words. Also there are other climate emulators that have a spatial element, so I'm not really sure what this sentence is trying to say.

Here we are saying that some simple climate models often achieve their improved runtime efficiency through  reductions in spatial detail, for example providing Global estimates of common climate metrics such as Mean Surface Temperature,  $CO_2$ concentration and Effective radiative forcing. We have modified the text to say:

"Simple climate models are extremely efficient although some only provide global estimates of climate metrics such as mean surface temperature, $CO_2$ concentration and Effective Radiative forcing."

Line 5: "general information" is vague

Agreed we have removed this and this sentence now reads:

"*Within the Intergovernmental Panel on Climate Change (IPCC) framework, understanding of the regional impacts of scenarios that include the most recent science is needed to allow targeted policy decisions to be made quickly.*"

~60: the overview of ongoing efforts is great. I think the machine learning models like Climatebench probably deserve a mention for completeness, since they have similar objectives

We agree and have now mentioned Climatebench in the manuscript saying:
*"the ClimateBench v1.0 (WatsonParris et al., 2022) benchmarks machine learning emulators that predict annual mean global distributions of temperature, diurnal temperature range and precipitation"*

66-68: I found this sentence confusing. "this type of input"? Not sure what input is being referenced

We have made this clearer by redrafting this text to say:

*"We use pattern-scaled climate variables instead of ESM output to drive our impacts model, because this approach offers a useful opportunity to more quickly derive impacts information from new scenarios. However, this does not imply that pattern-scaled climate variables should replace ESMs or ISIMIP bias-corrrected data but could provide a steer on which scenarios would be most useful for ESMs to run or which ones to bias-correct for use in more specialist impacts models."*

98-99: Choices affect the way patterns are selected? Do you just mean users choose the patterns? Also, please define Rose suite.

Yes the user can choose which patterns to include from which ESMs. We have clarified the text to say:

*"PRIME is a flexible framework, with ensemble members and patterns selected by the user and therefore dependent on their chosen application."*

Apologies for not defining what a Rose suite is in the manuscript, this is amended in the revised manuscript to say:

*"we are developing software to simplify running the PRIME framework using the choices presented here using Rose and Cylc (Oliver et al., 2018 and Cylc documentation 2024) - a group of utilities and specifications which provide a common way to manage the development and running of scientific application suites in both research and production environments. Rose and Cylc are used to ensure a consistent framework for managing and running meteorological and climate models, they are therefore ideally suited to this application."*

135: As stated repeatedly, part of the appeal of emulators is the low computational cost. However, the sampling done here is fairly sparse (Figure 3) and the reason given here is to "make the ensemble size manageable". I think it would be worth dedicating a bit of text to explain this discrepancy.

Thank you -this is a fair comment (no pun intended). The PRIME system is indeed intended to be fast in the sense that it is orders of magnitude faster than an ESM. The temporal and spatial detail of the land model means that PRIME is slower than FAIR itself as a global emulator, and the volume of data produced is substantial. As such PRIME sits between the two classes with the UKESM model achieving circa 2 years per day and FAIR running at approximately 11 years per second. PRIME runs approximately 20000 years per day using a moderate compute resource and applying limits to the number of runs at any one time.

Hence we still need to be mindful of run length and data volume. For uses where we expect substantial exploitation of results – e.g. if PRIME was to be used as a fast-response tool for new scenarios in AR7, then a much larger ensemble could still be fairly easily produced and data made available.

An additional appeal of emulators - including PRIME - is that they can run from global mean forcing, whereas to run an ESM requires many weeks of human effort to set up spatially-resolved forcing data - some of which is not available for novel scenarios from IAMs.

We have added the following text to Section 3 to explain this choice:

*"Figure 3 shows the selection of ensemble members from the full FaIR distribution of 2237 members; these 9 percentiles (0, 1, 5, 25, 50, 75, 95, 99 and 100\%) are chosen to explore the full range of global*

*temperature sensitivity, but make the data more manageable because it increases considerably when combined with the CMIP6 patterns (see Table S1 for a full list of those used) and run through JULES."*

136: "these are selected using one scenario so that scenarios can be compared against each other" is confusing

The temperature percentiles will be different for each scenario, so this means 1% will not be the same ensemble member for SSP5-8.5 as for example SSP1-2.6 or SSP5-3.4-OS. Unfortunately, this means if we choose different ensemble members for the 1% for each of the three scenarios it will be difficult to compare between scenarios. Selecting a single scenario for defining the ensemble members for each percentile makes it much easier to compare across scenarios. This is partly why we found it is useful to look at the joint temperature and $CO_2$ distribution for each scenario to convince ourselves this approach would work. We are actively considering our approach to ensemble selection across the temperature and $CO_2$ distributions for future versions of PRIME.

We have added the following text to better explain the use of a single scenario for selection of ensemble members.

*"We use a single scenario to define the ensemble member per percentile because each scenario will have different ensemble members for each percentile. For example, the 50th percentile ensemble member for SSP5-8.5 would not be the same ensemble member as the 50th percentile for SSP5-3.4-OS. We choose the same ensemble members for all scenarios to make the comparison between scenarios easier."*

147-148: Do the linear regressions include an intercept? If so, how is it treated?
Please also specify one realization of each model (i.e. not an ensemble and not all available)

Thank you for this comment, the linear regression intercepts are zero and the realization used for each CMIP6 ESM is in Table 1 of the supplementary information. We have added to the text referencing this table, to say that it includes which realization is used. The text has been updated to reflect this comment at the start of Section 2.2 (approximately line 165) in this latest revised version.

152: I think there needs to be a summary of this conversion from monthly to 3-hourly data. Not necessarily here, but somewhere.

The summary of the conversion from monthly to 3-hourly data is provided later in this section from line 190 onwards. We agree that this sentence would fit better later in this paragraph, where the weather generator is described, so we have now moved it to this location.

159: "this is the method currently used in PRIME" - it's not clear what is being referred to

This refers to the pattern scaling approach, to make this clearer the following text it now says:

*"Within PRIME we use patterns for all input variables required to run the JULES land-surface model. JULES tends to be less sensitive to some of the input variables that do not typically scale as well with temperature, such as wind speed, pressure and longwave downwelling radiation, so we can include them without introducing erroneous output changes (see Section 3.2). It should be noted that we generate*

*global patterns that include land and ocean but in this analysis, we focus on the patterns over land for running JULES and considering land impacts. However, it would be possible also to use the patterns over the ocean for relevant downstream applications."*

161: I thought the methods were described quite clearly up until paragraph, where it starts to get confusing. It's somewhat unclear what belongs to IMOGEN and what does not. I also don't understand why the diurnal cycle calculation could cause numerical instabilities (166) when there are no feedbacks. And what part of this is 3 hourly and where is the monthly temperature from the pattern scaling coming into play? Is there only one diurnal cycle for each month?

We have rewritten this section, making the steps from climate patterns to weather data to drive JULES clearer. It is the lack of the diurnal cycle in JULES that could cause instabilities. The text now says:

*"The spatial distribution of the monthly mean meteorology for each month of the transient simulation is reconstructed from the climate patterns multiplied by the global mean temperature change (see Section 2.1) superimposed on an observed monthly climatology. This is done by IMOGEN. In this version of PRIME, the observed monthly climatology was constructed from the daily meteorological data provided by the GSWP3-W5E5 dataset from the ISIMIP3a project (Frieler et al., 2023) for the period 1901--1930. This was regridded to a resolution of N48 with a 3.75° longitude grid size and a 2.5° latitude grid size.*

*In addition, the weather generator in IMOGEN (huntingford et al.,2010) is used to downscale the weather data from the monthly to hourly timestep, which is the temporal resolution used to drive JULES. This method is described in detail in (mathison et al.,2022). One limitation of this method is the lack of variability in the driving humidity, temperature and radiation at both the sub-daily and daily resolution. In the next version of PRIME we will develop the temporal downscaling meteorology so that it coherently includes the effects of, for example, clouds on the diurnal cycle of the weather data."*

164: Why are 1850-1889 anomalies added to 1901-1930 climatology? If it's observationally limited (i.e. no data prior to 1900), then it seems like the anomalies should also be relative to 1901-1930? Maybe you could at least demonstrate that this is a reasonable approximation, but it seems inconsistent. Also why did you decide to add it to an observational climatology instead of a modeled one?

The period 1901-1930 was a compromise between the pre-industrial (very limited observational data) and present day (lots of readily available observational data). We wanted an observational climatology where there was very little climate change, but where we had some (albeit quite limited) observational data. FaIR had a maximum global mean temperature change of 0.01 K by 1930 and less for the mean of the period 1901 to 1930 - this falls well within natural variability. We used the observed climate rather than a modelled climate so we could get the most realistic present day conditions and minimize differences between the model output and observed land surface.

Equation 3: I think that 3 should be an exponent - please check the equation

This equation has been updated.

194: typo - mode, not model

No this is correct and should say model

213: I don't think you mean "where ESMs have not been run" because the simulations exist

We do mean this because ultimately we want to use this framework to run scenarios that have not yet been run using ESMs, but in this paper we are showing that the framework is fit for purpose by running it using scenarios that are known and have previously been run using ESMs. We clarify this by modifying it to say:

*"In this section, we evaluate PRIME. In this context, that means that we show that the framework is 'fit for purpose' by testing it on scenarios where ESM simulations already exist. However, ultimately we want to use PRIME to produce land simulations for scenarios where ESMs have not been run "*

Figure 2 and S1. Caption should specify where the data are from. It says in Fig S1, but not in Fig 2. Could you also mention in the text again why you're subsampling? It hasn't been mentioned in several pages

I think the reviewer means Figure 3 and S1 as these are very similar plots showing the joint distribution from FaIR for $CO_2$ and temperature for the 3 scenarios covered with one scenario (SSP1-2.6) shown in Figure 3 and the other two scenarios (SSP5-3.4-OS and SSP5-8.5) shown in Figure S1. We have updated the caption of Figure 3 and Figure S1, so the details are now the same for these two figures.

*"Figure 3. Joint frequency distribution from the FaIR simulations of Temperature (TAS) and $CO_2$ concentration in 2100 for SSP1-2.6 emissions and the sub-selected percentiles (blue crosses) used to drive the JULES impacts model. Shades of green denote the density of points with individual histograms above and to the right of the main panel. 10% confidence intervals are shown by the contours."*

We also now restate that the FaIR distribution consists of 2237 ensemble members, so we sub-sample from this distribution to make the amount of data more manageable. For each FaIR ensemble member selected, we use 34 patterns from CMIP6 and we run JULES for each of these models. To run JULES we generate 3-hourly data for each of the 8 input variables. Currently we select 9 ensemble members using the temperature distribution, so for each scenario we have over 300 runs of JULES with all the output that generates.

*"Figure 3 shows the selection of ensemble members from the full FaIR distribution of 2237 members, these 9 percentiles (0, 1, 5, 25, 50, 75, 95, 99 and 100%) are chosen to explore the full range of global temperature sensitivity, but make the data more manageable because it increases considerably when combined with the CMIP6 patterns (see Table S1 for a full list of those used) and run through JULES."*

Figure 4: I don't think the differences should be on the same scale as the anomalies. For example, 4c just shows that most differences are less than 0.6 degrees, but grouping 0.01 to 0.6 together does not make sense when almost all values are less than 0.6. Plus the colors are faint (for the same reason) so they are hard to see. The figure captions (for 4 and accompanying SI figures) should also specify "anomalies" for them "predicted ensemble means"

Thank you for this comment, we agree the colours are faint in the difference plot and there is a balance to be found between being able to see the detail but also trying to show that these differences are small. As mentioned in the caption, the colour bar magnitude for the differences in this revision is the same as that

for the anomalies, in order to show that the prediction error is small compared to the change induced by the scenario. However we do accept that this does make the colours very faint and it is difficult to see the differences. We experimented with this, but found that those differences for temperature are faint no matter what range we use in the colour bar. In any case, we have adjusted the colour bars for all the variables, this includes Figure 4 and the associated supplementary figures (Figure S2-S9) to try and make the differences clearer but also not overstate these, because in most cases these differences are quite small.

The figure captions of Figure 4 and S2 to S9 have also been updated to say "Evaluation of the pattern predicted ensemble mean anomalies…"

Figure 5: I think adding a ratio column (MAE / IQR) would better make the point.

We have added an MAE/IQR column to Figure 5 and the equivalent plots in the Supplementary information.

S15: All colors in all of the related figure are shown from light to dark, except pressure in S15. Although it's not wrong, I would recommend reversing the colors for clarity

We have modified the plots showing the IQR/MAE for Pressure (i.e. plots equivalent to Figure 5) to go from light to dark. The modified plots are in the supplementary and are S11 and S15.

307: This sentence and several sentences in this paragraph are a little wordy and could be edited for clarity. For example, the reference to Burton et al does not seem to fit well in the context, and "actual values" on line 315 is vague.

Agree that the reference to Burton et al is a little separate to where we discuss the variables that are important for JULES, so this has been removed in the revised version and also added CMIP6 to the sentence highlighted, so it now reads "CMIP6 values", this occurs at line 346 in the latest revision. We have also edited this paragraph to make it "less wordy".

345: What does it mean for the pattern scaling technique to be well understood by the literature?

We agree that the meaning of this sentence is unclear so have removed this reference to the literature as it is enough to say that the method is well understood.

366: I'm a little confused here - you would not expect the values to be identical but "hope for a similar spread". I agree it seems to work impressively well, but there is some disconnect between saying there are several differences (e.g. only using one land model) and yet expecting similar spread. The rest of the paper is about emulating spread when using many models, so why would you expect the spread here to be similar when only using one model? A bit of clarifying text would be helpful.

Yes, you are correct. It is true that we would not expect the spread to be exactly the same (in fact if we did then this would imply that different land models add no value!). Although here we have an additional

source of spread by the sampling of FAIR sensitivity ranges for all climate patterns. What is important is that the use of a single land model here does not overly restrict the output and negate the benefits of being able to sample climate sensitivity and climate patterns fully. We have revised the text to clarify this:

*"It is important to check that the use of a single land model here does not overly restrict the output and negate the benefits of being able to sample climate sensitivity and climate patterns fully, so while we would not expect PRIME values to be identical to CMIP values for these JULES outputs, we check that the use of a single land model does not result in too narrow a range of outcomes."*

Figure 9: The caption says "median and uncertainty ranges" but I don't see that in the figure. I am confused about what the lines are - can you clarify in the text and caption? Are they for different pattern scaling from each of the different climate models?

Thank you for this comment, this was an oversight. Each of the lines in Figure 9 represent output from each PRIME ensemble member. In an earlier version of this plot, we only showed the median and uncertainty ranges but during the peer-review process we have revised this to show each ensemble member. We have revised the caption to reflect this.

*"Figure 9. Maps of net ecosystem production (top) and tree fraction (bottom) with timeseries showing PRIME output for each ensemble member for each study region: Amazon, Siberia, India and the USA (labelled) for SSP1-2.6 between 1850–2100."*

469: You mention that local land-atmosphere feedbacks are not included, but that seems secondary to me given the fact that the land model produces carbon-cycle relevant parameters like GPP that do not feed back into FaIR. As mentioned above, I think that qualification should be added.

Please see the response to the query in the section - Reviewer general comments, where we explain that we are using simulations which are "concentration-driven" which means that even for carbon-cycle ESMs, these models also do not allow this feedback onto atmospheric $CO_2$ to operate. This means that the comparison made between PRIME and CMIP6 is still a fair one. However, the authors accept that the one-way coupling of the PRIME framework means that the land -surface cannot then have an effect on the emissions driving FaIR and have added text to the manuscript in the discussion at the end of Section 5.0 (around line 484 in the latest revised version) to explain that this is a future ambition but is not possible in PRIME in its current form.